# Opioid antagonism modulates wanting-related frontostriatal connectivity

Alexander Soutschek[1]*, Susanna C Weber[2], Thorsten Kahnt[3], Boris B Quednow[4,5], Philippe N Tobler[2,5]

[1]Department of Psychology, Ludwig Maximilian University, Munich, Germany; [2]Zurich Center for Neuroeconomics, Department of Economics, University of Zurich, Zürich, Switzerland; [3]Department of Neurology, Northwestern University Feinberg School of Medicine, Chicago, United States; [4]Experimental and Clinical Pharmacopsychology, Department of Psychiatry, Psychotherapy and Psychosomatics, Psychiatric Hospital, University of Zurich, Zurich, Switzerland; [5]Neuroscience Center Zurich, University of Zurich and Swiss Federal Institute of Technology Zurich, Zürich, Switzerland

**Abstract** Theoretical accounts distinguish between motivational ('wanting') and hedonic ('liking') dimensions of rewards. Previous animal and human research linked wanting and liking to anatomically and neurochemically distinct brain mechanisms, but it remains unknown how the different brain regions and neurotransmitter systems interact in processing distinct reward dimensions. Here, we assessed how pharmacological manipulations of opioid and dopamine receptor activation modulate the neural processing of wanting and liking in humans in a randomized, placebo-controlled, double-blind clinical trial. Reducing opioid receptor activation with naltrexone selectively reduced wanting of rewards, which on a neural level was reflected by stronger coupling between dorsolateral prefrontal cortex and the striatum under naltrexone compared with placebo. In contrast, reducing dopaminergic neurotransmission with amisulpride revealed no robust effects on behavior or neural activity. Our findings thus provide insights into how opioid receptors mediate neural connectivity related to specifically motivational, not hedonic, aspects of rewards.

*For correspondence:
alexander.soutschek@psy.lmu.de

## Editor's evaluation

The authors measured the effects of the opioid receptor antagonist naltrexone (50mg), and the dopamine D2/3 antagonist amisulpiride (400mg), on self-reported reward wanting vs. liking and functional connectivity between the prefrontal cortex and striatum (using functional magnetic resonance imaging) in healthy human participants, using a between-subjects design. Naltrexone led to lower wanting, but not liking, and these changes were associated with greater frontostriatal connectivity. Amisulpiride also tended to increase connectivity on wanting trials but did not affect either wanting or liking scores. The results raise the possibility that both opioid and dopamine transmission influence reward wanting, with the former more closely related to conscious processes. This manuscript will be of broad interest to neuroscientists interested in the brain mechanisms underlying reward processing, both at the circuit and molecular levels.

## Introduction

Rewards are central for goal-directed behavior as they induce approach behavior toward valued outcomes (*Schultz, 2015*). Theoretical models distinguish between behavioral dimensions of rewards, such as the motivational drive to obtain rewards ('wanting') versus the hedonic pleasure associated with reward consumption ('liking'), whereby 'wanting' and 'liking' refer to preconscious, rather than

conscious, mental states (*Berridge, 1996*; *Berridge and Kringelbach, 2015*; *Berridge et al., 2009*). Dysfunctions in wanting and liking of rewards belong to the core symptoms to addiction, which can be conceptualized as a wanting-dominated state with deficits in switching to liked non-drug rewards (*Berridge and Robinson, 2016*; *Berridge et al., 2009*). It is thus important to obtain a better understanding of the human brain mechanisms underlying wanting and liking. Previous animal research suggested that wanting and liking relate to dissociable neurochemical mechanisms (*Berridge and Kringelbach, 2015*; *Berridge and Valenstein, 1991*): Dopaminergic activity is thought to modulate the wanting component of rewards, but not liking (*Berridge and Valenstein, 1991*). In contrast, the opioidergic system has later been associated with both wanting and liking (*Berridge and Kringelbach, 2015*). Human studies support the hypothesized link between dopaminergic activation and cue-triggered wanting (*Hebart and Gläscher, 2015*; *Soutschek et al., 2020b*; *Weber et al., 2016*) as well as the motivation to work for rewards (*Cawley et al., 2013*; *Chong et al., 2015*; *Korb et al., 2020*; *Skvortsova et al., 2017*; *Soutschek et al., 2020a*; *Venugopalan et al., 2011*; *Westbrook et al., 2020*; *Zénon et al., 2016*). Noteworthy, two of these studies suggest that dopamine changes only experimental measures of wanting (motivation to work for rewards), but not self-reported wanting ratings (*Korb et al., 2020*; *Venugopalan et al., 2011*).

Consistent with animal findings, pharmacological manipulations of the opioid system affected both wanting and liking aspects of rewards in humans (*Buchel et al., 2018*; *Chelnokova et al., 2014*; *Eikemo et al., 2016*). Less is known, however, about the neuroanatomical basis of human wanting and liking. Both dopaminergic and opioidergic neurons project to reward circuits in the striatum as well as to the prefrontal cortex (*Delay-Goyet et al., 1987*; *Lidow et al., 1991*), and recent neuroimaging findings suggest that these regions indeed play a role in processing of wanting and liking (*Weber et al., 2018*). In particular, the ventral striatum appears to encode the currently behaviorally relevant reward dimension and dynamically switch functional connectivity with wanting- and liking-encoding prefrontal regions (*Weber et al., 2018*; such frontostriatal gating constitutes one possibility how dopamine [and by extension opioids] can affect functions involving prefrontal cortex and the striatum; *Cools, 2011*). This is also in line with the view that wanting and liking are preferentially represented in partially distinct subregions of the striatum (which are in turn connected to different input and output regions; *Peciña, 2008*). Thus, processing of wanting and liking appears to be dissociable on both a neurochemical and an anatomical basis. However, it remains unknown how pharmacological and connectivity-related brain mechanisms interact. We investigated whether frontostriatal connectivity related to motivational and hedonic judgements is modulated by dissociable neurotransmitter systems.

To test this hypothesis, the current study investigated the impact of pharmacologically manipulating dopaminergic and opioidergic systems on the neural processing of wanting and liking information. This study was part of a larger project investigating also the roles of opioidergic and dopaminergic activity for reward impulsivity (*Weber et al., 2016*). Here, we administered a task that allows distinguishing between wanting and liking dimensions of valued goods and assessed how pharmacologically reducing dopaminergic (using the dopamine antagonist amisulpride) or opioidergic neurotransmission (with the opioid antagonist naltrexone) changes frontostriatal connectivity related to parametric wanting and liking judgements. We hypothesized that, on a behavioral level, reducing opioid receptor activation will reduce both wanting and liking (*Buchel et al., 2018*; *Chelnokova et al., 2014*; *Eikemo et al., 2016*), whereas reduced dopaminergic neurotransmission will selectively affect wanting (*Cawley et al., 2013*; *Korb et al., 2020*; *Venugopalan et al., 2011*). We further hypothesized that on a neural level the behavioral effects of the pharmacological manipulations are mirrored by changes in frontostriatal connectivity related to wanting and liking information (*Weber et al., 2018*). In particular, we expected that frontostriatal connectivity during wanting judgements is modulated by opioidergic and dopaminergic activation (as these neurotransmitters have been related to the processing of wanting), whereas frontostriatal connectivity during liking judgements should be reduced after reduction of opioidergic neurotransmission.

## Results

### Opioid antagonism reduces wanting ratings

We analyzed the data of healthy young volunteers who rated how much they wanted or liked everyday items in the MRI scanner. We collected wanting and liking ratings for all items in the MRI scanner twice, once before (pre-test session) and once after (post-test session) participants played a game on the computer where they won or lost 50% of the items (in order to have equal numbers of won and lost items for the statistical analysis). This allowed us to assess whether participants behaviorally distinguished between wanting and liking ratings, because based on our previous findings we expected that winning and losing items has dissociable effects on wanting and liking (*Weber et al., 2018*). Participants actually received the won items at the end of the experiment (i.e., after the post-test session). We therefore selected everyday items (e.g., batteries or candles – for the full list of items, see *Weber et al., 2018*) that should be both wanted and liked by the majority of our participants from the Zurich student population. To test the impact of pharmacologically manipulating dopaminergic and opioidergic receptor activation on wanting and liking, participants received either naltrexone (*N* = 37), amisulpride (*N* = 40), or placebo (*N* = 39) prior to performing the task in the scanner.

First, we performed a sanity check whether participants distinguished between wanting and liking ratings by assessing the impact of winning versus losing items on wanting and liking ratings in the post-test session. As recommended for pre-test/post-test designs (*Dugard and Todman, 1995*), we regressed ratings in the post-test session on item-specific pre-test ratings. Moreover, we included predictors for *Judgement* (wanting versus liking), *Item type* (lost versus won), and the interaction terms. Contrary to our previous study (*Weber et al., 2018*), we observed no significant *Judgement× Item* type interaction, $\beta$ = 0.50, $t$(111) = 1.61, p = 0.11, which does not replicate our previous result that winning versus losing items has dissociable effects on wanting versus liking of the items (*Figure 1B* and *Table 1*). However, separate analyses for wanting and liking ratings revealed no significant difference in wanting ratings between won and lost items, $\beta$ = 0.29, $t$(115) = 0.64, p = 0.52, whereas liking was more strongly reduced for lost than for won items, $\beta$ = 01.84, $t$(115) = 3.16, p = 0.002, with the latter effect replicating our result of decreased liking of lost versus won items (*Weber et al., 2018*).

Next, we assessed the impact of reducing dopamine and opioid receptor activity on wanting and liking judgements. We analyzed ratings (pre- and post-test) with predictors for *Amisulpride* (versus placebo), *Naltrexone* (versus placebo), *Judgement*, *Session* (pre-test versus post-test), and the interaction terms. This analysis provided evidence that reducing opioid neurotransmission differentially affected wanting and liking ratings, *Naltrexone × Judgement*, $\beta$ = 7.02, $t$(125) = 2.36, p = 0.02, while we observed no significant effects for amisulpride, $\beta$ = 3.79, $t$(126) = 1.30, p = 0.20 (*Figure 1C* and *Table 2*). Judgement type-specific analyses suggested that wanting ratings were significantly reduced under naltrexone (mean = 4.5, standard deviation [sd] = 1.0) relative to placebo (mean = 4.9, sd = 1.0), $\beta$ = –13.85, $t$(115) = 2.12, p = 0.04, Cohen's *d* = 0.47, whereas amisulpride (mean = 4.9, sd = 1.0) showed no significant effects on wanting ratings relative to placebo, $\beta$ = –1.39, $t$(116) = 0.22, p = 0.83, Cohen's *d* = 0.05. Neither naltrexone (mean = 5.2, sd = 0.9) nor amisulpride (mean = 5.4, sd = 0.8) showed significant effects on liking relative to placebo (mean = 5.2, sd = 0.8), for both $t$ < 1.17, p > 0.24, Cohen's *d* < 0.27. Taken together, our findings provide evidence for involvement of opioidergic neurotransmission in wanting judgements.

Amisulpride can show both pre-synaptic and post-synaptic effects depending on the administered dose. To control for the possibility that the effective dose of amisulpride might vary between participants due to differences in body weight, we added the predictor body weight (as well as its interactions with all other factors) to the above reported regression model. While the *Naltrexone × Judgement* interaction remained significant, $\beta$ = 7.99, $t$(124) = 2.69, p = 0.008, there were still no significant amisulpride effects, all $t$ < 1.46, all p > 0.14. There was thus no evidence for dose-dependent effects of amisulpride on wanting or liking ratings.

To assess the robustness of these findings, we conducted also a non-hierarchical analysis of pharmacological effects on wanting and liking ratings using the mean wanting and liking ratings across all items, determined separately for each participant and session (pre-test versus post-test). The analysis of wanting ratings replicated the significant main effect of naltrexone versus placebo, $t$(128) = 2.16, p = 0.03, while amisulpride showed no significant effect on mean wanting ratings, $t$(128) = 0.23, p = 0.82. Mean liking ratings were neither affected by naltrexone, $t$(126) = 0.03, p = 0.98, nor amisulpride,

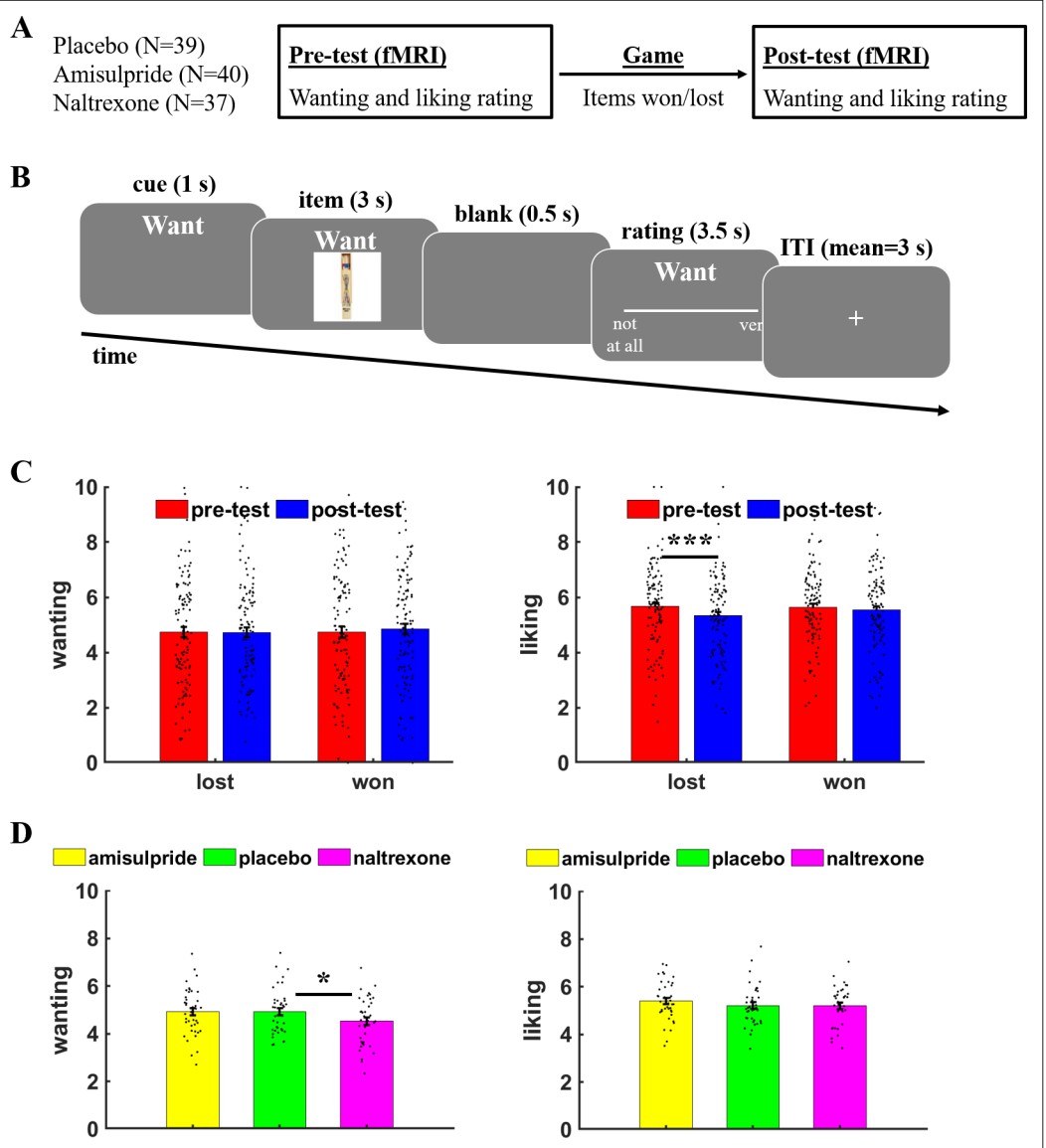

**Figure 1.** Task procedure and behavioral results. (**A**) Participants rated in the MRI scanner how much they wanted or liked objects before (pre-test) or after (post-test) they won or lost these items in a game between the scanning sessions. (**B**) On each trial, a cue indicated whether a wanting or liking rating was required, followed by the presentation of the current object (here: a pick-up sticks game). Participants had to rate how much they wanted or liked the presented object within 3.5 s, then the next trial started after a variable inter-trial interval (mean = 3 s). (**C**) Liking ratings were significantly reduced for objects that were lost relative to won in the gamble, while wanting ratings did not significantly differ between lost versus won items. (**D**) The opioid antagonist naltrexone significantly reduced wanting ratings relative to placebo, while liking ratings were unaffected by naltrexone or the dopamine antagonist amisulpride. For illustration purposes, participant-specific mean wanting/liking ratings are plotted on a scale from 0 to 10, while the statistical analyses are conducted on the participant- and item-specific wanting and liking ratings. Error bars indicate standard error of the mean, black dots represent individual data points. *p < 0.05, ***p < 0.001.

$t(126) = 1.22$, $p = 0.23$. Thus, also the non-hierarchical analysis of aggregated mean data provided no evidence for significant amisulpride effects.

## Opioid antagonism reduces wanting-related frontostriatal connectivity

Next, we investigated the neural mechanisms underlying the impact of opioid antagonism on wanting. Following the procedures from our previous study (**Weber et al., 2018**), we first determined the

**Table 1.** Results of mixed general linear model 1 (MGLM-1) on wanting and liking ratings in the post-test as function of *Judgement* (wanting versus liking), *Item type* (lost versus won), and *Pre-test* ratings.

Standard errors of the mean (SE) are in brackets.

|  | Beta (SE) | t-Value | df | p-Value |
|---|---|---|---|---|
| Intercept | 0.21 (0.66) | 0.32 | 119 | 0.75 |
| Judgement | −2.27 (0.64) | 3.54 | 125 | <0.001 |
| Item type | 0.07 (1.10) | 0.07 | 98 | 0.95 |
| Pre-test | 79.87 (0.50) | 158.30 | 124 | <0.001 |
| Judgement × Item type | 2.01 (1.24) | 1.62 | 109 | 0.11 |
| Judgement × Pre-test | 2.41 (0.61) | 3.93 | 289 | <0.001 |
| Item type × Pre-test | −1.62 (0.71) | 2.26 | 133 | 0.03 |
| Judgement × Item type × Pre-test | 0.91 (0.90) | 1.01 | 288 | 0.31 |

neural correlates of wanting and liking by computing GLM-1 in which onset regressors for wanting and liking judgements were modulated by non-orthogonalized parametric modulators for wanting and liking ratings. Wanting ratings (independently of the required judgement type) correlated with activation in ventromedial prefrontal cortex (VMPFC; $z = 7.32$, whole-brain FWE-corrected, p < 0.001, peak = [0 44–7]), dorsolateral prefrontal cortex (DLPFC; $z = 6.63$, whole-brain FWE-corrected, p < 0.001, peak = [−21 38 44]), and posterior cingulate cortex (PCC; $z = 5.29$, whole-brain FWE-corrected, p = 0.002, peak = [−3 −37 38]) (*Figure 2A* and *Table 3*). Liking ratings correlated with BOLD signal changes in more posterior parts of PCC ($z = 4.37$, whole-brain FWE-corrected, p = 0.02, peak = [−9 −64 38]) (*Figure 2B* and *Table 4*). Moreover, we also replicated our previous finding that liking ratings correlate with activity in orbitofrontal cortex (OFC) when applying small-volume correction (SVC; anatomical mask for the OFC based on the wfupickatlas; $z = 3.20$, small-volume FWE-corrected, p = 0.046, peak = [−21 50–4]). Together, these data replicate our previous findings that wanting and liking are correlated with activation in VMPFC and OFC, respectively. However, we observed no significant effects of naltrexone or amisulpride (relative to placebo) on these neural representations of wanting

**Table 2.** Results for mixed general linear model 2 (MGLM-2) assessing drug effects on wanting and liking ratings as function of *Drug* (amisulpride versus placebo and naltrexone versus placebo), *Judgement* (wanting versus liking), and *Session* (pre-test versus post-test).

Standard errors of the mean (SE) are in brackets.

|  | Beta (SE) | t-Value | df | p-Value |
|---|---|---|---|---|
| Intercept | 3.62 (5.50) | 0.66 | 98 | 0.51 |
| Amisulpride | 2.37 (5.09) | 0.47 | 114 | 0.64 |
| Naltrexone | −6.80 (5.19) | 1.31 | 114 | 0.19 |
| Judgement | 4.38 (2.08) | 2.11 | 125 | 0.04 |
| Session | −3.30 (1.98) | 1.67 | 1612 | 0.10 |
| Amisulpride × Judgement | 3.79 (2.92) | 1.30 | 126 | 0.20 |
| Naltrexone × Judgement | 7.02 (2.98) | 2.36 | 125 | 0.02 |
| Amisulpride × Session | 0.02 (2.79) | 0.00 | 1629 | 0.99 |
| Naltrexone × Session | 2.70 (2.84) | 0.95 | 1618 | 0.34 |
| Judgement × Session | −1.09 (1.97) | 0.55 | 1845 | 0.58 |
| Amisulpride × Judgement × Session | −1.50 (2.78) | 0.54 | 1856 | 0.59 |
| Naltrexone × Judgement × Session | −2.95 (2.83) | 1.04 | 1853 | 0.30 |

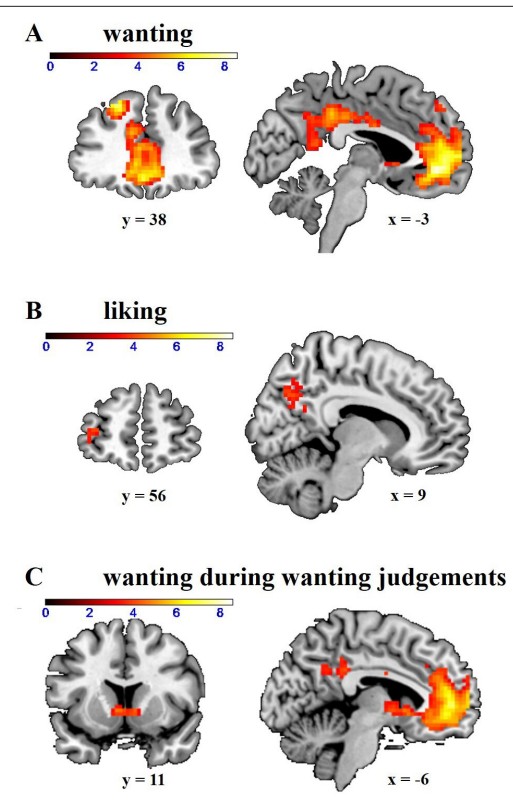

**Figure 2.** Neural correlates of (**A**) wanting and (**B**) liking independently of behavioral relevance. Wanting correlated with activation in dorsolateral prefrontal cortex (DLPFC), ventromedial prefrontal cortex (VMPFC), and posterior cingulate cortex (PCC) (whole-brain FWE-corrected). Liking correlated with activation in dorsal PCC (whole-brain FWE-corrected) and orbitofrontal cortex (small-volume FWE-corrected). (**C**) Wanting ratings significantly correlated with activation in the striatum during wanting judgements (small-volume FWE-corrected). Images are thresholded at p < 0.001 uncorrected.

or liking in these regions (or at the whole-brain level), even at lenient statistical thresholds (p < 0.001 uncorrected, cluster size >20 voxels).

GLM-1 revealed no significant wanting- or liking-related striatal activation, which may appear surprising given the canonical role of the striatum for reward processing (*Bartra et al., 2013*). However, this might be due to the fact that the parametric modulators for wanting and liking only explained unique variance (regressors were not orthogonalized), while striatal activation might be shared by wanting and liking. To test this, we computed two further GLMs, one (GLM-3) where we orthogonalized liking with respect to wanting (such that the regressor for wanting explained the variance shared by wanting and liking) and one where we orthogonalized wanting with respect to liking (GLM-4). In GLM-3, we observed bilateral wanting-related activation in the striatum ($z$ = 7.09, whole-brain FWE-corrected, p < 0.001, peak = [−9 11 −1]), PCC ($z$ = 12.22, whole-brain FWE-corrected, p < 0.001, peak = [0 −28 25]), VMPFC ($z$ = 11.13, whole-brain FWE-corrected, p < 0.001, peak = [0 47 −7]), posterior parietal cortex ($z$ = 8.42, whole-brain FWE-corrected, p < 0.001, peak = [−45 −67 35]), and DLPFC ($z$ = 5.59, whole-brain FWE-corrected, p < 0.001, peak = [24 32 47]). Likewise, in GLM-4 liking ratings (including the variance shared with wanting) correlated with activation in striatum ($z$ = 6.19, whole-brain FWE-corrected, p < 0.001, peak = [−9 14 −1]), PCC ($z$ = 10.95, whole-brain FWE-corrected, p < 0.001, peak = [0 −31 35]), DLPFC ($z$ = 7.77, whole-brain FWE-corrected, p < 0.001, peak = [−18 32 50]), VMPFC ($z$ = 7.69, whole-brain FWE-corrected, p < 0.001, peak = [−3 50 −4]), and posterior parietal cortex ($z$ = 6.13, whole-brain FWE-corrected, p < 0.001, peak = [−45 −67 35]). Thus, both wanting and liking correlated with activation in regions belonging to the neural reward system. However, also in the GLMs with orthogonalized parametric modulators, we observed no effects of naltrexone or amisulpride (relative to placebo) on activations related to wanting (GLM-3) or liking (GLM-4) ratings even at lenient statistical thresholds (p < 0.001 uncorrected, cluster size >20 voxels).

Previous research showed that wanting-related prefrontal activation is functionally coupled with the ventral striatum depending on the behavioral relevance of wanting judgements (*Weber et al., 2018*). Consistent with our previous finding, striatal activation was significantly correlated with wanting ratings when those were behaviorally relevant (wanting ratings on wanting trials in GLM-2), $z$ = 4.46, p = 0.003, peak = [−6 11 −1], small-volume FWE-corrected with anatomical mask for the striatum (*Figure 2C*). We note that wanting-related striatal activation did not survive FWE correction at the whole-brain peak level (p = 0.14, although it did survive whole-brain FWE correction at the cluster level, p < 0.001), such that this effect appears to be somewhat weaker than in our previous study (*Weber et al., 2018*).

We next assessed whether wanting-related prefrontal regions are functionally connected with the striatum by conducting a psychophysiological interaction (PPI) analysis with the striatum as seed region. To test whether the pharmacological manipulations changed the functional connectivity of the

**Table 3.** Anatomical locations and MNI coordinates of the peak activations correlating with wanting ratings in general linear model 1 (GLM-1).

We report activations surviving whole-brain FWE correction at peak level (p < 0.05). Hem = Hemisphere (L = left, R = right); BA = Brodmann area.

| | | | MNI coordinates | | | | |
|---|---|---|---|---|---|---|---|
| Region | Hem | BA | X | Y | Z | k | Z |
| VMPFC | R/L | 10 | 0 | 44 | –7 | 439 | 7.32 |
| DLPFC | L | 8 | –21 | 38 | 44 | 39 | 6.63 |
| | L | 8 | –33 | 23 | 44 | 4 | 5.11 |
| PCC | L | 23 | -3 | –37 | 38 | 39 | 5.29 |
| | R/L | 23 | 0 | –13 | 35 | 1 | 4.79 |
| Anterior cingulate cortex | R | 32 | 6 | 35 | 11 | 3 | 5.27 |
| Frontopolar cortex | L | 10 | –12 | 65 | 20 | 1 | 4.82 |

striatum with specifically wanting-related brain regions, we applied SVC within a mask of significant wanting-correlated voxels in DLPFC and VMPFC in GLM-1 (thresholded with FWE at peak level, $k$ = 478). On wanting trials, we observed enhanced functional coupling between striatum and DLPFC as a function of increasing wanting ratings (wanting ratings on wanting trials: $z$ = 3.87, small-volume FWE-corrected, p = 0.02, peak = [–21 41 41]), as to be expected given that both the DLPFC and the striatum showed significant wanting-related activity. Moreover, DLPFC-striatum connectivity on wanting trials was stronger for wanting than for liking ratings (wanting > liking ratings on wanting trials: $z$ = 3.64, small-volume FWE-corrected, p = 0.04, peak = [–21 41 41]) (*Figure 3A*). The wanting-dependent DLPFC-striatum coupling is consistent with previous findings that connectivity between the stratum and prefrontal correlates of wanting depends on whether wanting judgements are behaviorally relevant (*Weber et al., 2018*).

Next, we tested how our pharmacological manipulation changed functional connectivity between the striatum and wanting-related cortical regions. Compared with placebo, naltrexone increased DLPFC-striatum coupling for wanting relative to liking ratings on wanting trials (naltrexone > placebo for wanting > liking ratings on wanting trials: $z$ = 3.81, small-volume FWE-corrected, p = 0.02, peak = [–18 35 38]) (*Figure 3B*). Moreover, the impact of naltrexone on DLPFC-striatum connectivity was significantly stronger on wanting than on liking trials, ((wanting > liking ratings)$_{\text{wanting trials}}$ > (wanting > liking ratings)$_{\text{liking trials}}$-related connectivity in the naltrexone relative to the placebo group: $z$ = 3.55, small-volume FWE-corrected, p = 0.05, peak = [–18 35 38]). Thus, the effects of naltrexone on fronto-striatal connectivity were specific for wanting judgements. We observed no further regions showing significantly reduced wanting-related connectivity under naltrexone relative to placebo, and we also observed no significant differences between amisulpride and placebo as well as naltrexone and amisulpride. Thus, reducing opioid neurotransmission strengthened the functional connection between the striatum and prefrontal cortex when wanting judgements were behaviorally relevant.

**Table 4.** Anatomical locations and MNI coordinates of the peak activations correlating with liking ratings in general linear model 1 (GLM-1).

We report activations surviving whole-brain FWE correction at peak level (p < 0.05). Hem = Hemisphere (L = left, R = right); BA = Brodmann area.

| | | | MNI coordinates | | | | |
|---|---|---|---|---|---|---|---|
| Region | Hem | BA | X | Y | Z | k | Z |
| Dorsal PCC | L | 31 | –9 | –64 | 38 | 1 | 4.87 |

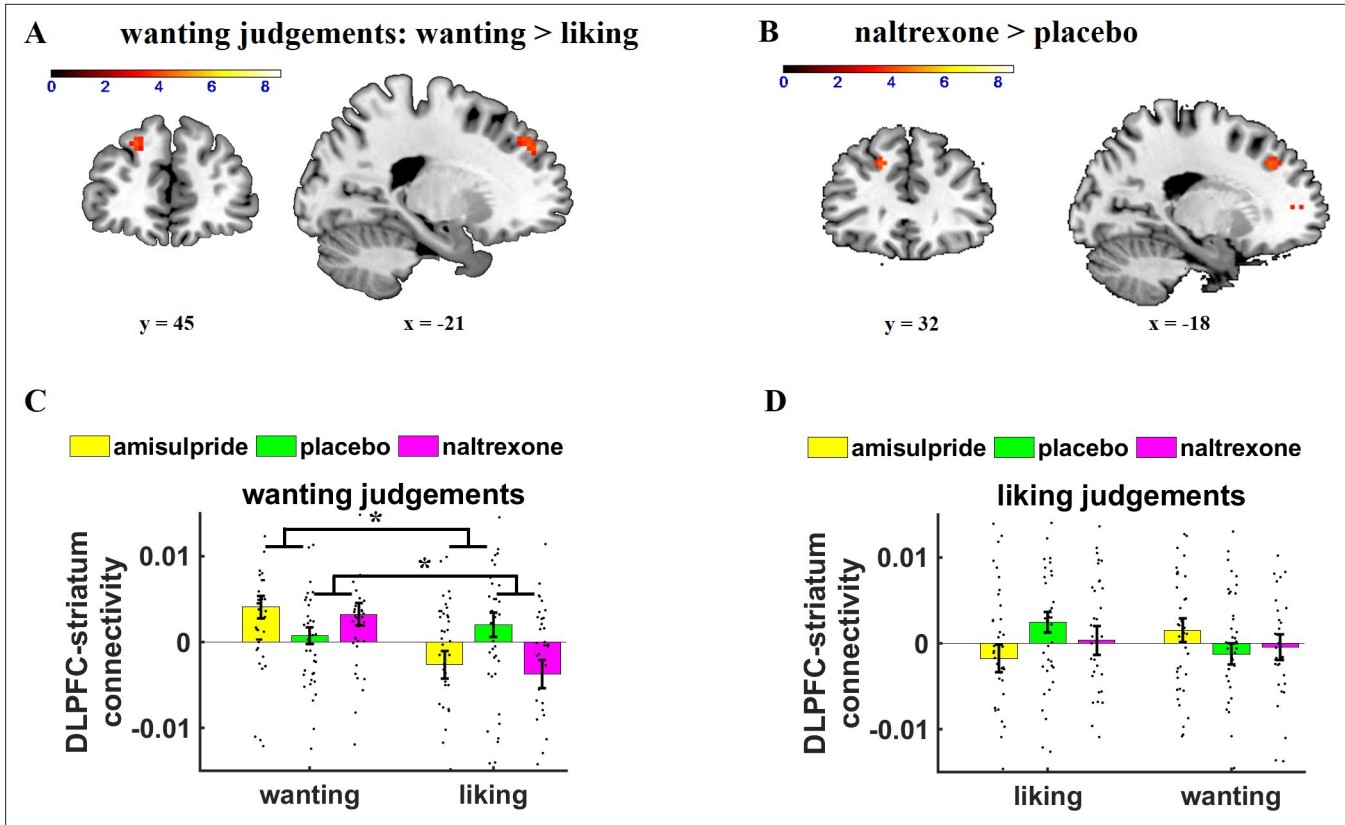

**Figure 3.** Effects of Judgement type and drug on parametric striatal connectivity. (**A**) On wanting trials (collapsed across drug groups), dorsolateral prefrontal cortex (DLPFC)-striatum connectivity was enhanced for wanting relative to liking aspects of rewards (image thresholded at p < 0.001 uncorrected). (**B**) Wanting-related DLPFC-striatum coupling was significantly stronger under naltrexone compared with placebo (image thresholded at p < 0.001 uncorrected). (**C, D**) Extracted parameter estimates for DLPFC (as defined by the significant cluster in general linear model 1 [GLM-1]), separately for wanting and liking judgements. (**C**) If wanting judgements were behaviorally relevant, naltrexone increased wanting relative to liking-related DLPFC-striatum connectivity. (**D**) No significant drug effects on DLPFC-striatum connectivity were observed on liking trials. Error bars indicate standard error of the mean, black dots represent individual data points. *p < 0.05.

The observed drug effects on the neural level raise the question whether the impact of naltrexone on behavioral wanting ratings can statistically be explained by its influence on DLPFC-striatum connectivity. For this purpose, we extracted parameter estimates from the significant DLPFC cluster for the (wanting > liking ratings)$_{\text{wanting trials}}$-contrast and tested whether the impact of naltrexone on wanting ratings is mediated by the naltrexone effects on DLPFC-striatum connectivity (using quasi-Bayesian confidence intervals as implemented in the mediation package for R; *Tingley et al., 2014*). However, the test of the significance of the indirect path (which is the decisive criterion for the presence of a mediation effect; *Zhao et al., 2010*) showed no significant effect, ACME = –0.18, p = 0.06. Thus, the data do not provide sufficient evidence that naltrexone effects on behavior can be explained with modulation of frontostriatal connectivity. We also note that the main effect of naltrexone on wanting ratings remained significant when controlling for DLPFC-striatum connectivity, $\beta$ = –8.48, t(76) = 2.54, p = 0.01, while DLPFC-striatum connectivity was stronger for more highly wanted items in the placebo group, $\beta$ = 586, t(73) = 2.30, p = 0.02. We must therefore be careful with interpreting the naltrexone effects on brain connectivity as the potential cause for the behavioral drug effects.

For completeness, despite having observed no significant drug effects on liking in the behavioral analysis, we also performed whole-brain analyses assessing which brain regions show enhanced functional connectivity as function of liking ratings. On liking trials, connectivity with the striatum was stronger for liking than for wanting ratings in OFC, (z = 3.16, small-volume FWE-corrected, p = 0.05, peak = [–33 50–7]), replicating our previous findings. However, no brain regions showed significant effects of naltrexone or amisulpride (relative to placebo) on liking-related connectivity with the striatum even at low, exploratory statistical thresholds (p < 0.001, cluster size >20 voxels).

To assess the robustness of the naltrexone effects on wanting-related DLPFC-striatum connectivity, we extracted parameter estimates from the significant wanting-related DLPFC cluster in GLM-1 and regressed the parameter estimates on predictors for Drug, Judgement, Relevance, and the interaction terms (using the lmer function in R). The significant *Naltrexone× Judgement × Relevance* interaction, $\beta$ = 6.0e-03, $t$(452) = 2.46, p = 0.01, replicated the finding that naltrexone had dissociable effects on wanting and liking as a function of the behavioral relevance of these reward components. We also observed a significant Amisulpride × *Relevance* interaction, $\beta$ = 4.1e-03, $t$(452) = 2.42, p = 0.02. Separate analyses for wanting and liking judgements revealed that in wanting trials DLPFC-striatum connectivity was enhanced for wanting compared with liking ratings under both naltrexone, *Naltrexone× Relevance* interaction, $\beta$ = 5.6e-03, $t$(226) = 3.19, p = 0.002, and amisulpride, Amisulpride × *Relevance* interaction, $\beta$ = 4.1e-03, $t$(226) = 2.35, p = 0.02 (*Figure 3C*). In contrast, liking judgements showed no significant effects of naltrexone or amisulpride relative to placebo, all $t$ < 1.34, all p > 0.18 (*Figure 3D*). This result supports our findings based on SVC according to which naltrexone increases wanting-related relative to liking-related DLPFC-striatum connectivity on wanting trials, and hints at a similar function for amisulpride (though this was not evident in the SVC-based analysis).

## Discussion

In animal models, the wanting and liking dimensions of rewards are processed by partly distinct brain regions and neurotransmitter systems, but in humans it remained unclear so far how opioidergic and dopaminergic systems orchestrate the processing of wanting and liking. The current findings provide evidence for dissociable roles of opioidergic neurotransmission in processing the two dimensions of rewards on both a behavioral and a neural level. Behaviorally, lowering opioidergic activation with naltrexone selectively reduced wanting, not liking, ratings for non-consumable goods. On a neural level, this reduction in wanting was reflected by changes in DLPFC-striatum connectivity: When wanting judgements were required, DLPFC-striatum connectivity was significantly stronger for the behaviorally relevant wanting ratings than for the irrelevant liking dimension of rewards.

Importantly, wanting-related functional coupling between DLPFC and striatum was significantly stronger under naltrexone than under placebo. This is consistent with recent findings relating opioid receptor inhibition with increased connectivity between the prefrontal control system and reward circuits (*Elton et al., 2019*; *Lim et al., 2019*) and suggesting prefrontal kappa opioid receptors to mediate the impact of naltrexone on drug craving in alcohol use disorder (*de Laat et al., 2019*). Through corticostriatal loops, the striatum receives input from several cortical regions and can prioritize processing of behaviorally relevant information (*Frank, 2011*). DLPFC provides inhibitory input to the striatum and was shown to reduce wanting-related activation in the striatum (*Dong et al., 2020*; *Koob and Volkow, 2010*), consistent with the importance of frontostriatal loops for self-control (*van den Bos et al., 2014*). According to this view, the observed positive relationship between wanting ratings and DLPFC-striatum connectivity might indicate that DLPFC exerts top-down control over striatal representations of wanting predominantly for highly wanted items, whereas there may be less need for inhibitory top-down control for less desired goods (*van den Bos et al., 2014*). By strengthening DLPFC-striatum connectivity, naltrexone enhances top-down processes predominantly for highly wanted items, which would explain why lower wanting under naltrexone is associated with stronger DLPFC-striatum connectivity even though in the placebo group functional DLPFC-striatum coupling is increased for highly wanted items. In any case, because we observed no significant brain-behavior mediation effect (p = 0.06), the stronger DLPFC-striatum connectivity under naltrexone should not be interpreted as the cause for the changes in behavioral ratings under naltrexone. It is further worth noting that in the current study we observed significant effects predominantly in the left hemisphere. In the literature on frontostriatal connectivity during reward processing, both left- and right-lateralized effects were reported (*van den Bos et al., 2014*; *van den Bos et al., 2015*; *Yuan et al., 2017*). We therefore do not make any claims regarding whether this result pattern represents just a power issue or a truly lateralized effect.

Contrary to our hypotheses, we did not observe effects of naltrexone on liking or amisulpride effects on wanting. Interestingly, however, a recent study observed no influences of opioid and dopamine antagonists on self-report wanting and liking ratings but only on experimental measures of these reward dimensions (*Korb et al., 2020*). In fact, previous studies reporting effects of dopaminergic manipulations on wanting operationalized wanting with experimental measures rather than self-report

(*Soutschek et al., 2020a*; *Soutschek et al., 2020b*; *Weber et al., 2016*), while studies using self-report ratings observed no or only weak effects of pharmacological manipulations (*Case et al., 2016*; *Ellingsen et al., 2014*; *Løseth et al., 2019*). In line with our previous study (*Weber et al., 2018*), we had decided to use self-report ratings to avoid the issue that implicit measures of liking such as face muscle activity are more open to alternative interpretations (*Pool et al., 2016*), but we acknowledge that implicit measures might be more sensitive to pharmacological interventions. Moreover, given that in our study participants had to provide liking ratings without being able to actually consume or handle the items, our measurements might have been less sensitive than those of other studies assessing the liking of consumed rewards (*Buchel et al., 2018*; *Chelnokova et al., 2014*; *Eikemo et al., 2016*). It is also worth noting that it has recently been suggested that amisulpride shows, if any, only weak effects on BOLD signal changes in the reward system (*Grimm et al., 2020*). We note though that in the same sample of participants amisulpride showed significant effects on tasks for cue reactivity and delay discounting (*Weber et al., 2016*), which were administered 2.5 hr after drug intake (while the rating task started 1 hr after drug intake). Due to this difference in timing, it is thus not possible to decide whether the different amisulpride effects on these tasks can be explained by different sensitivities of these tasks to dopaminergic manipulations or by the time course of amisulpride effects. In any case, one should thus be careful with interpreting these unexpected null findings as being inconsistent with previous pharmacological results manipulating dopaminergic activity with different compounds than amisulpride.

Interestingly, the ROI analysis provided some evidence for amisulpride effects on wanting at the neural level, as amisulpride increased wanting-related DLPFC-striatum connectivity, similar to the findings for naltrexone. However, the impact of amisulpride on wanting-related frontrostriatal connectivity (gating) needs to be interpreted with caution, given the lack of significant amisulpride effects on behavior.

Our findings have important implications for clinical research, given that dysfunctions in wanting and liking are prevalent in several psychiatric disorders. Substance use disorders, for example, are characterized by increased wanting of drugs as reflected in craving symptoms (*Berridge, 2012*; *Edwards, 2016*), and craving has been linked to impairments in prefrontal top-down control over the striatum (*Feil et al., 2010*). Naltrexone is approved in several countries for the treatment of alcohol use disorder (*Krystal et al., 2001*; *Srisurapanont and Jarusuraisin, 2005*) and opioid dependence (*Johansson et al., 2006*) and was shown to reduce relapse risk and craving specifically in alcohol use disorder. Consistent with the view that naltrexone reduces the salience of drug cues by strengthening prefrontal activation (*Courtney et al., 2016*), we speculate that the beneficial effects of naltrexone on alcohol use and craving might be explained by increased top-down control of DLPFC over striatal wanting signals as a consequence of opioid receptor inhibition (but see *Nestor et al., 2017*). Our results may thus improve the understanding of neural mechanisms underlying pharmacological treatments of dysfunctional wanting in substance use disorders.

Several limitations are worth mentioning. First, we did not assess wanting and linking prior to drug administration, such that we cannot control for potential baseline differences in wanting and liking between drug groups. Thus, it remains possible that the non-significant effects of amisulpride on wanting and of naltrexone on liking are caused by such pre-existing baseline differences, or that the sample size was not sufficient to detect these effects in a between-subject design. We also note that the doses for amisulpride and naltrexone might not have been pharmacologically equivalent. In fact, while 50 mg naltrexone produces 95% μ-opioid receptor occupancy (*Weerts et al., 2008*), 400 mg amisulpride leads to a lower dopamine receptor occupancy of 85% (*Lako et al., 2013*), which might be a further reason for why naltrexone showed stronger effects on behavior and brain activation than amisulpride. Lastly, while high doses of amisulpride (≥400 mg) reduce postsynaptic dopaminergic signaling, lower doses of amisulpride increase dopaminergic activity via presynaptic mechanisms (*Schoemaker et al., 1997*), but higher doses may also increase signaling at D1 receptors and thereby counteract the inhibitory effects on D2 neurotransmission. As the effective dose of amisulpride might differ between participants (*Sescousse et al., 2016*), one might argue that presynaptic and postsynaptic effects of dopamine might have canceled out across participants, leading to the observed null effect of amisulpride on behavior on the group level. However, contrary to this view, we observed no significant amisulpride effects even when controlling for body weight as proxy for effective dose, and we note that in previous studies we had observed effects of 400 mg amisulpirde on behavior (*Burke*

*et al., 2018; Soutschek et al., 2017*) and multivariate neural data (*Kahnt et al., 2015*). It seems thus unlikely that the null effects of amisulpride on the rating task can be explained solely by the chosen dosage.

Taken together, our findings deepen our understanding of the neurochemical mechanisms mediating the impact of wanting of rewards on behavior. Opioid receptors are involved in the modulation of the strength of inhibitory prefrontal input to the striatum encoding the behavioral relevance of the wanting dimension of rewards. These insights into the interactions between neuroanatomical and neurochemical brain mechanisms implementing wanting-driven approach behavior advance our understanding of the mechanisms underlying pharmacological treatments of substance use disorders.

## Materials and methods

### Participants

A total of 121 healthy volunteers (58 females; $M_{age}$ = 21.8 years, range = 18–30), recruited via email from the internal pool of the Laboratory for Social and Neural Systems Research (which includes mainly students from the University of Zurich and ETH Zurich), participated in the study. According to power analysis assuming the effect size (Cohen's $d$ = 0.65) from a previous study in our lab on the impact of amisulpride on value representations in the neural reward system (*Kahnt et al., 2015*), 38 participants per group allow detecting a significant effect (alpha = 5%) with a power of 80%. The goal of the power analysis was to optimize the sample size for finding drug effects on neural reward signals. However, given the differences between the current study design and the study by *Kahnt et al., 2015*, we note that the power might not have been optimal for all statistical tests in the current investigation (e.g., drug effects on explicit ratings or functional connectivity). Three participants were excluded from the analysis due to response omissions in more than 30% of all trials in the rating task (see below), two further participants were excluded because of excessive head movement (>5 mm in one of the six head motion parameters) in the scanner. Thus, the final sample comprised 116 participants (placebo: $N$ = 39; naltrexone: $N$ = 37; amisulpride: $N$ = 40). Drug groups were matched with regard to age (p = 0.40), sex (p = 0.34), years of education (p = 0.45), and BMI (p = 0.29). Participants were screened prior to participation for exclusion criteria including history of brain disease or injury, surgery to the head or heart, and neurological or psychiatric diseases (including alcohol use disorder, depression, schizophrenia, bipolar disorders, claustrophobia, or Parkinson symptoms) via paper-pencil questionnaires. Further exclusion criteria were a severe medical disease such as diabetes, cancer, insufficiency of liver or kidneys, acute hepatitis, high or low blood pressure, any cardiovascular incidences, epilepsy, pregnancy or breastfeeding, past use of opiates or other drugs that may interact with amisulpride or naltrexone (such as stimulants). A qualitative drug urine screening test (M-10/5-DT, Diagnostik Nord, Schwerin, Germany) was performed to rule out illicit drug use prior the test session (amphetamines, barbiturates, buprenorphine, benzodiazepines, cannabis, cocaine, MDMA, methadone, and morphine/opiates). All participants provided written informed consent. For their participation, they received 40 Swiss francs per hour. The study was approved by the Ethics Committee of the Canton of Zurich and was part of a larger project where we investigated also pharmacological effects on Pavlovian-to-instrumental transfer and delay discounting (published in *Weber et al., 2016*). These tasks were administered after the rating task reported in the current manuscript (3 hr after drug intake). The larger project (though not the rating task) was preregistered on https://www.clinicaltrials.gov/ (NCT02557984).

### Procedure

Participants received a pill containing either placebo ($N$ = 40), 400 mg amisulpride ($N$ = 41) or 50 mg naltrexone ($N$ = 40) in a randomized and double-blind manner 3 hr before performance of the experimental tasks. Amisulpride is a selective dopamine D2/D3 receptor antagonist, whereas naltrexone is an unspecific opioid receptor antagonist that acts primarily on the μ- and $\kappa$-opioid receptors, with lesser and more variable effects on δ-opioid receptors (*Rosenzweig et al., 2002; Weerts et al., 2008*). We asked participants to fast for 6 hr before arrival at the lab. One hour after drug intake, participants started the wanting/liking rating task (see below) in the MRI scanner, which took approximately 90 min. The (first) peak in plasma concentration for amisulpride is after 60 min (*Rosenzweig et al., 2002*), whereas for naltrexone the peak is after 120 min (*Verebey et al., 1976*), such that

participants performed the rating task around peaks in average plasma concentration. After task completion, participants answered post-experimental questionnaires, which probed whether they thought they had received a drug or placebo, and measured their mood (one rating was not recorded in the placebo group). We determined amisulpride and naltrexone blood plasma levels immediately before and after the behavioral tasks with high-performance liquid chromatography–mass spectrometry in order to control for the pharmacokinetics of the drugs.

### Task design

Participants performed a task in which they had to rate how much they wanted or liked 40 non-consumable everyday items (*Weber et al., 2018*). Before performing the rating task in the scanner, we familiarized participants with the items by physically presenting all items to them. The rating task was implemented in Matlab (The MathWorks, Natick, MA) and the Cogent 2000 toolbox. We asked participants to rate each item according to how much they wanted to have it, as well as how much they liked the item at that moment. In each trial, participants first saw a cue indicating the type of rating (wanting or liking) (1 s), followed by an image of the item (3 s), and finally the rating screen (3.5 s). Ratings were provided on a continuous scale using a trackball. Trials were separated by a variable intertrial interval (mean 3 s). Each item was rated twice for wanting and twice for liking, resulting in 160 trials split into four runs before the game (pre-test) and four runs after the game (post-test). Between the pre-test and post-test experimental sessions, participants played a game inside the scanner in which they could win the items. The game consisted of a perceptual task in which participants had to indicate whether the item was presented to the left or the right of the midpoint of the screen. Participants won items that they classified correctly. The difficulty of the game was calibrated such that participants won and lost 50% of the items.

### MRI data acquisition and preprocessing

Whole-brain scanning was performed with a Philips Achieva 3T whole-body MRI scanner equipped with an eight-channel head coil (Philips, Amsterdam, The Netherlands). For each of the eight scanning runs, 227 T2*-weighted whole-brain EPI images were acquired in ascending order. Each volume consisted of 33 transverse axial slices, using field of view 192 mm × 192 mm × 108 mm, slice thickness 2.6 mm, 0.7 mm gap, in-plane resolution 2 mm × 2 mm, matrix 96 × 96, repetition time (TR) 2000 ms, echo time (TE) 25 ms, flip angle 80°. Additionally, a T1-weighted turbo field echo structural image was acquired for each participant with the same angulation as applied to the functional scans (181 slices, field of view 256 mm × 256 mm × 181 mm, slice thickness 1 mm, no gap, in-plane resolution 1 mm × 1 mm, matrix 256 × 256, TR 8.4 ms, TE 3.89 ms, flip angle 8°).

Preprocessing was performed with SPM 12 (https://www.fil.ion.ucl.ac.uk/spm/). The functional images of each participant were motion corrected, unwarped, slice-timing corrected (temporally corrected to the middle image), and co-registered to the anatomical image. Following segmentation, we spatially normalized the data into standard MNI space. Finally, data were smoothed with a 6 mm FWHM Gaussian kernel and high-pass filtered (filter cutoff = 128 s).

### Behavioral data analysis

Behavioral data in the rating task were analyzed with mixed general linear models (MGLMs) using the lme4 package in R. The alpha threshold was set to 5% (two-tailed). Degrees of freedom and p-values were computed using the Satterthwaite approximation with the lmerTest package. To replicate our previous findings that winning versus losing items has dissociable effects on wanting and liking ratings, we regressed item-specific ratings in the post-test session on fixed-effect predictors for *Judgement* (wanting versus liking), *Item type* (lost versus won), z-transformed item-specific ratings in the pre-test, and all interaction terms (MGLM-1). All these predictors were also modeled as random slopes in addition to participant-specific random intercepts. We also performed separate analyses for wanting and liking ratings (MGLM-2) where post-test item-specific ratings were predicted by *Item type* and *Pre-test ratings*.

To assess drug effects on wanting and liking ratings, we regressed session- and item-specific ratings on fixed-effect predictors for *Drug* (amisulpride versus placebo and naltrexone versus placebo), *Judgement*, *Session* (pre-test versus post-test), and the interaction effects (MGLM-3). All fixed effects varying on the individual level (i.e., *Judgement*, *Session*, and *Judgement × Session*) were

also modeled as random effects in addition to participant-specific intercepts. Again, we performed separate analyses for wanting and liking (MGLM-4), which were identical to MGLM-3 but left out all predictors for *Judgement*.

## MRI data analysis

To investigate drug effects on neural activation related to wanting and liking ratings, we computed two GLMs, following previous procedures (*Weber et al., 2018*). GLM-1 included an onset regressor for the presentation of the current item and the rating bar (duration = 7 s). This onset regressor was modulated by three mean-centered parametric modulators, that is, mean session-specific and item-specific wanting and liking ratings as well as decision times (to control for choice difficulty). The mean item-specific correlation between wanting and liking ratings was $r = 0.71$. We did not orthogonalize the parametric modulators, such that the results for the regressors reflect the unique variance explained by wanting or liking ratings. A separate regressor modeled all items for which no session- and item-specific value could be computed due to response omissions. GLM-2 was identical to GLM-1 with the only difference that it included separate onset regressors for wanting and liking trials. In GLM-2, the onset regressors for wanting and liking trials were modulated by parametric modulators for both wanting and liking ratings, which allowed assessing judgement-specific (e.g., wanting ratings on wanting trials) and judgement-unspecific (e.g., liking ratings on wanting trials) neural correlates of wanting and linking (*Weber et al., 2018*). Finally, we computed two further models, one where the liking regressor in GLM-1 was orthogonalized with respect to wanting (such that the regressor for wanting contained the variance shared by wanting and liking; GLM-3) and one where wanting was orthogonalized with respect to liking (GLM-4). In all models, the regressors were convolved with the canonical hemodynamic response function in SPM. We also added six movement (three translation and three rotation) parameters as covariates of no interest.

For statistical analysis, we first computed the following participant-specific contrasts: For GLM-1, we computed parametric contrasts for wanting ratings and liking ratings (independently of judgement type) in GLM-1. For the second-level analysis, we entered the contrast images from all participants in a between-participant, random effects analysis to obtain statistical parametric maps. First, we investigated the neural correlates of wanting and liking independently of administered drug and conducted whole-brain second-level analyses using one-sample *t*-tests. To assess drug effects, we employed second-level independent *t*-tests for naltrexone versus placebo as well as amisulpride versus placebo. For these analyses, we report results that survive whole-brain family-wise error corrections at the peak level. In the figures, we set the individual voxel threshold to p < 0.001 with a minimal cluster extent of $k \geq 20$ voxels. Results are reported using the MNI coordinate system.

## PPI analysis

To examine how our pharmacological manipulations modulated the frontostriatal connectivity of wanting and liking, we conducted a whole-brain PPI analysis with the striatum as seed region. We defined the seed region by building a sphere (diameter = 6 mm) around the coordinates of wanting-related striatum activation in GLM-2 (MNI coordinates: $x = -6$, $y = 11$, $z = -1$). To create the regressors for the PPI analysis, we first extracted the average time course from the seed region for each individual participant (physiological regressor). We then multiplied the physiological regressor with psychological regressors for (i) wanting ratings on wanting trials, (ii) liking ratings on wanting trials, (iii) liking ratings on liking trials, and (iv) wanting ratings on liking trials. Next, we computed a GLM (PPI-1) that included the interaction terms, the physiological regressor, and the psychological regressors. We also added separate onset regressors for wanting and liking trials as well as movement parameters as regressors of no interest. For the statistical analysis, we computed contrasts for wanting ratings on wanting trials, wanting > liking ratings on wanting trials, liking ratings on liking trials, and finally liking > wanting ratings on liking trials. We submitted these contrasts to a second-level analysis to yield statistical parametric maps with a one-sample *t*-test. Because GLM-1 revealed wanting ratings to correlate with activation in DLPFC and VMPFC, we tested whether the striatum seed region shows functional connectivity with DLPFC and VMPFC. For this purpose, we applied SVC with a mask that included the significant wanting-correlated voxels in bilateral VMPFC and DLPFC from GLM-1 (thresholded with FWE correction at the voxel level). Additionally, we also performed exploratory whole-brain analyses.

## Acknowledgements

We thank Karl Treiber for expert support with data collection as well as Beatrice Beck Schimmer for medical support.Funding and financial disclosure: PNT received funding from the Swiss National Science Foundation (Grants 10001C_188878, 100019_176016, and 100014_165884) and from the Velux Foundation (Grant 981). AS received an Emmy Noether fellowship (SO 1636/2-1) from the German Research Foundation.

## Additional information

### Competing interests

Thorsten Kahnt: Reviewing editor, *eLife*. The other authors declare that no competing interests exist.

### Funding

| Funder | Grant reference number | Author |
| --- | --- | --- |
| Schweizerischer Nationalfonds zur Förderung der Wissenschaftlichen Forschung | Grants 10001C_188878 and 100014_165884 | Philippe N Tobler |
| Schweizerischer Nationalfonds zur Förderung der Wissenschaftlichen Forschung | 100019_176016 | Philippe N Tobler |
| Velux Stiftung | 981 | Philippe N Tobler |
| Deutsche Forschungsgemeinschaft | SO 1636/2-1 | Alexander Soutschek |

The funders had no role in study design, data collection and interpretation, or the decision to submit the work for publication.

### Author contributions

Alexander Soutschek, Formal analysis, Writing - original draft; Susanna C Weber, Conceptualization, Formal analysis, Investigation, Writing - review and editing; Thorsten Kahnt, Boris B Quednow, Conceptualization, Writing - review and editing; Philippe N Tobler, Conceptualization, Funding acquisition, Supervision, Writing - original draft

### Author ORCIDs

Alexander Soutschek http://orcid.org/0000-0001-8438-7721
Thorsten Kahnt http://orcid.org/0000-0002-3575-2670
Philippe N Tobler http://orcid.org/0000-0002-4915-9448

### Ethics

Clinical trial registration https://www.clinicaltrials.gov/ (NCT02557984).
Human subjects: All participants provided written informed consent. The study was approved by the ethics committee of the canton of Zurich (KEK-ZH-NR2012-0347).

### Decision letter and Author response

Decision letter https://doi.org/10.7554/eLife.71077.sa1
Author response https://doi.org/10.7554/eLife.71077.sa2

## Additional files

### Supplementary files

• Transparent reporting form

## Data availability

The behavioral data that support the findings of this study are available on Open Science Framework https://osf.io/6cevt/.

The following dataset was generated:

| Author(s) | Year | Dataset title | Dataset URL | Database and Identifier |
| --- | --- | --- | --- | --- |
| Soutschek A | 2021 | Wanting_liking_pharma_fmri | https://osf.io/6cevt/ | Open Science Framework, 6cevt |

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
