## [Editor Report]

The authors measured the effects of the opioid receptor antagonist naltrexone (50mg), and the dopamine D2/3 antagonist amisulpiride (400mg), on self-reported reward wanting vs. liking and functional connectivity between the prefrontal cortex and striatum (using functional magnetic resonance imaging) in healthy human participants, using a between-subjects design. Naltrexone led to lower wanting, but not liking, and these changes were associated with greater frontostriatal connectivity. Amisulpiride also tended to increase connectivity on wanting trials but did not affect either wanting or liking scores. The results raise the possibility that both opioid and dopamine transmission influence reward wanting, with the former more closely related to conscious processes. This manuscript will be of broad interest to neuroscientists interested in the brain mechanisms underlying reward processing, both at the circuit and molecular levels.

---

## [Decision Letter]

**Decision letter after peer review:**

Thank you for submitting your article "Opioid antagonism reduces wanting by strengthening frontostriatal connectivity" for consideration by *eLife*. Your article has been reviewed by 3 peer reviewers, and the evaluation has been overseen by a Reviewing Editor and Michael Frank as the Senior Editor. The following individual involved in review of your submission has agreed to reveal their identity: Guillaume Sescousse (Reviewer #2).

The reviewers have discussed their reviews with one another, and the Reviewing Editor has drafted the text below to help you prepare a revised submission.

The reviewers agreed that this was an ambitious study and that with appropriate analysis and revisions the results are likely to interest a broad audience due to the translational interest in how dopamine and opioid systems affect liking and wanting of rewards across species. They felt that the observation that blocking opioid receptors specifically reduces the wanting of rewards, possibly via modulating frontostriatal coupling, is a potentially important result; and that the methodology underpinning the results was generally appropriate and solid. However, they also commented that the current version of the manuscript lacks much necessary methodological information, and that a substantial rewrite including considerable additional information and some further analysis is necessary.

We are therefore issuing a revise and resubmit decision, but with no guarantee that the revised submission will be accepted as the reviewers' confidence in the findings will in large part rest on your responses to their queries.

Essential revisions:

Introduction

1) The paper appears to rest heavily on a single neuroimaging paper from the same group, and some prior findings from that study are replicated, albeit with a very small effect size. It neglects much of the rest of the relevant literature however, e.g. in the numerous unreferenced claims in the Introduction.

Methods

2) Please specify the urine tox screen used. Please explain how participants were recruited and identified as being free of exclusion criteria. For example, was a formal, semi-structured clinical interview for psychiatric disorders conducted?

3) The authors write that wanting and liking ratings were elicited twice (before and after fMRI), yet analyses appear to indicate that these judgements were also elicited during fMRI? A number of other details about the task are missing, such as the rationale for selection of items, whether these were actually won or hypothetically won, why the 'difficulty' was set to 50%, and why only explicit ratings were included.

4) The authors report that they did not orthogonalize the parametric modulators modelling the wanting and liking ratings. While this makes sense as a way to ensure that each modulator captures unique variance, presumably wanting and liking ratings are tightly correlated. Was that the case? Could the authors provide some details about the average correlation between the ratings?

5) The reviewers appreciated the power analysis. However some details are missing so that the numbers provided can be fully appreciated by the readers. In particular, the study cited as a reference for effect sizes (Kahnt et al., 2015) used a different task and different analyses, which makes it unclear how the authors were able to derive a relevant effect size estimate. Also, since there are many effects that can be tested in the present study (i.e. various behavioral and brain-related effects and correlations), it is important to report explicitly for which of these effects sample size was optimized.

6) The reviewers also appreciated the pre-registration on clinicaltrials.gov. However again some details seem to be missing. The pre-registration mentions a PIT task and a delay discounting task, but no rating task. This discrepancy requires explanation. Furthermore, the pre-registration implies that this study was embedded in a larger project involving more tasks than the one reported here. It is important to report this information explicitly, and explain how the rating task took place in the context of this larger protocol.

Results

7) The reviewers queried the statistics presented in the analysis of the behavioural data. In Figure 1, the amisulpride group shows clearly elevated ratings of wanting and liking, at a magnitude similar to the effect of naltrexone on wanting, but the difference between amisulpride and placebo is reported as non-significant for liking ratings. The statistics reported do not appear to match the means – e.g. in panel D. The difference between naltrexone and placebo for wanting looks similar (with even larger between-subject variance), so it seems surprising that the effect of amisulpride on liking ratings is non-significant. Can the authors verify there is not a mistake? It also seems from Figure 1D that amisulpride might have a main (increasing) effect on ratings: have the authors checked for such a main effect?

8) The authors should explain how the degrees of freedom reported for the contrasts were determined – they often do not appear to match either the number of participants or the number of observations.

9) The description of the Judgement × Item type interaction as "marginally significant" is inappropriate. A quick check with http://statcheck.io/ shows that the two-tailed p-value is 0.11, so the one-tailed p-value is 0.055. Following recent recommendations in the field (e.g. Gibbs and Gibbs, 2015; Otte et al., 2021), especially in a context where some scientists have called for lowering the statistical threshold of p<0.05 for significance (Benjamin et al., 2017), the authors should report this result as non-significant, and/or substantiate it with a complementary statistical approach.

10) On a related point, on p5 a p-value of 0.52 cannot support a claim of the form "wanting ratings did not differ between won and lost items". Such a p-value > 0.05 merely suggests that the data at hand don't provide sufficient evidence to reject the null. Bayesian statistics could potentially be used to make the sort of claims made by the authors. The same issue applies on p. 6: "amisulpride did not change wanting ratings relative to placebo, β = -1.39, t(116) = 0.22, p = 0.83".

11) Ideally one would like to establish a causal mechanism from the present results: is reduced wanting following opioid receptor blockade mediated by a direct action of naltrexone on frontostriatal connectivity (as suggested in the title of this submission)? Have the authors have considered performing a mediation-style analysis?

12) On p9 it would informative to present the striatal activation observed for the correlation with relevant wanting ratings. The statistics are reported for only one unilateral peak voxel, so it is hard to assess how robust this result is. This is particularly important since this result is supposed to be a replication of the authors' previous work (Weber et al., 2018).

13) The authors note that DLPFC-striatum connectivity was associated with self-reported wanting and that naltrexone increased this connectivity further. Since naltrexone also decreased self-reported wanting, are these results contradictory? This needs clarification.

Discussion

14) The authors used an intermediate dose of amisulpride (400 mg), which has been suggested to exert a mixture of pre- and post-synaptic effects depending on the participant, with pre-synaptic blockage likely to increase D1 receptor activation. Therefore any group-wise effect may cancel out across participants (see e.g. van der Schaaf et al., 2012; Sescousse et al., 2016; Eisenegger et al., 2014). Could this be a reason for the limited effects of amisulpride observed in the present study?

15) Some of the key results appear to be unilateral, i.e. specifically in the left hemisphere (striatal activity scaling with wanting ratings and DLPFC-striatum connectivity). Such a unilateral effect could reflect limited statistical power and fragile results, but also a truly lateralized effect. It would be worth commenting on this in the Discussion.

16) Based on previous literature it is surprising that the striatum was not part of the neural correlates of wanting and liking (Figure 2) – striatal activation is only identified when the ratings are restricted to the relevant ones. Yet, based on studies like Barta et al. (2013) or Lebreton et al. (2009) that show that the striatum is at the core of a robust and automatic valuation system, one might expect activity in this region to correlated with ratings even when these are not explicitly required. Could the authors comment on this?

*Reviewer #2:*

There is still an ongoing debate regarding the exact role of these neurotransmitters in the brain, and this study is very timely given that this debate has been mostly fueled by animal work. The present study builds on previous work from the authors (Weber et al., 2018), replicating some of their fMRI results, and adding a pharmacological manipulation. Using a between-subject, placebo-controlled, double-bling protocol, the authors show that blocking opioid receptors with naltrexone specifically reduces wanting of rewards, while blocking dopamine D2 receptors with amisulpride did not have noticeable effects. Importantly, the authors show that this pharmacological effect was accompanied by an increase in frontostriatal coupling. These results complement the previous study of the authors and strengthen their hypothesis of a frontostriatal gating of motivational – but no hedonic – aspects of reward processing.

This study has a variety a strengths in my opinion. The authors provide some evidence (even if sometimes limited) that some of their previous behavioral and fMRI results replicate. Sample sizes are relatively large for a pharmaco-fMRI study, and the methods are generally sound and well-justified, providing confidence that conclusions are well-supported by the data. The study was preregistered, which also increases faith in the results.

Nonetheless, some aspects could be improved. In particular, some statistical procedures would gain in being clarified or strengthened. Some important information is also sometimes missing, regarding methods and embedding of this study in a larger project. Finally, several points could be discussed in more detail in the Discussion, especially regarding striatal activity and drug dosage.

*Reviewer #3:*

This labor-intensive study tested contributions of dopamine and opioid systems to reward wanting vs. liking in healthy humans. Strengths include an impressive sample size for a single-center fMRI / pharmacological challenge study (N=116), carefully constructed hypotheses with suitable methods and statistical analyses, and an interesting question. The primary limitation was the use of a selective D2/3 receptor antagonist and dose that likely has complex effects, partially decreasing post-synaptic D2/3 transmission, increasing dopamine release (via blocked autoreceptors), and increasing D1 receptor transmission.

[Editors' note: further revisions were suggested prior to acceptance, as described below.]

Thank you for resubmitting your work entitled "Opioid antagonism modulates wanting-related frontostriatal connectivity" for further consideration by *eLife*. Your revised article has been reviewed by 3 peer reviewers and the evaluation has been overseen by Michael Frank as the Senior Editor, and a Reviewing Editor.

The reviewers and editors feel that the manuscript has been improved but there are several remaining major issues that need to be addressed, as outlined below. Please note that if these issues are not satisfactorily addressed in your revised submission then unfortunately we will not be able to consider the manuscript further, as it is not editorial practice to issue multiple revise resubmit decisions at *eLife*.

1) The most important issue is that there remains a discrepancy between similar effect sizes of naltrexone/amisulpride on wanting (albeit in opposite directions) and the corresponding pattern of P-values obtained from the hierarchical analysis. A similar issue is also present in relation to liking ratings. The authors need to explore this discrepancy in considerably more detail and resolve it, as follows:

a) Conduct a non-hierarchical analysis using the mean ratings for wanting and liking (in two separate models, one for wanting, one for liking). The reason for this is that it appears from the data depicted in Figure 1D that amisulpride may increase wanting and also liking (where the effect may actually be even greater). The reviewers noted that for liking ratings the mean difference is ~0.5 points and the SDs are actually lower than for wanting ratings at 1.0/1.2, which is suggestive of a larger effect than the effect of naltrexone on wanting which is significant in the hierarchical model. It would also be useful to provide the standardised effect sizes (Cohen's d) for the 4 comparisons against placebo (2 for wanting, 2 for liking).

b) Assuming that the above analyses using the mean ratings provide a discrepant pattern of significance to the hierarchical analysis, this then needs to be investigated thoroughly and explained in the manuscript, both for wanting and for liking ratings. The authors need to dig into the data carefully and figure out why this discrepancy arises. For example, if amisulpride makes participants more variable in their ratings (or naltrexone make them more consistent), this would be important for interpretation. Or perhaps some assumptions of the hierarchical model have been broken? Or perhaps the covariance structure requires amending? Or perhaps the model did not converge? Without resolution of this discrepancy it will not be possible to consider the manuscript further.

2) The authors now report the magnitude of the correlation between wanting and liking ratings, which is unsurprisingly high (r = 0.71). Since they have not serially orthogonalized the parametric regressors in the main analysis, this means that much of the variance of these ratings is simply removed. For this reason it is not clear how much we can infer from the non-significant drug effects, considering that these were assessed using only a fraction of the ratings variance, which may result in an insensitive analysis. Therefore further analyses are required here to substantiate the conclusion that there were no drug effects (as reported on the top of p10 – it is assumed that currently this refers to the model *without* serial orthogonalisation, although this should be stated explicitly for clarity).

The authors do provide some results from an analysis using serial orthogonalisation, with liking orthogonalised against wanting (p9), yielding the expected striatal activation for the parametric effect of wanting (which then carries the shared variance), which is reassuring – as they note this suggests that the striatal signal is substantively affected by the collinearity between wanting and liking ratings. Please additionally report the drug effects in this analysis. The authors should also report the effects from an analysis in which the serial orthogonalisation is performed in the alternate order (i.e. wanting against liking, such that the liking regressor now carries the shared variance), including both the main parametric effect (this time of liking) and drug effects.

3) It is helpful that the authors added the information that previous data from the same study were published in a 2016 paper by Weber et al. Oddly they do not mention the results of that paper, even in the discussion of the (apparent – see point 3 below) non-significant effects of amisulpride. The 2016 findings are highly relevant, since the amisulpride group was found to suppress cue-based responding and reward impulsivity. Similar results, but weaker, were reported for naltrexone. Both groups also reported lower mood than the placebo group.

The authors explain that the PIT and delay discounting tasks were completed after the end of scanning, i.e. after 60 minutes absorption time + 90 minutes fmri rating task = minimum 2.5 hours after drug administration. Hence, it seems highly relevant for the interpretation of the present data that, in the exact same participants, the same dose of amisulpride reported to show a null during 1-2.5 hours, showed what are (presumably) expected effects after 2.5 hours. Therefore it is necessary to mention this prior publication from the same study in the Introduction, and discuss the results, especially with respect to dose timing, in the Discussion.

---

## [Author Response]

Essential revisions:Introduction1) The paper appears to rest heavily on a single neuroimaging paper from the same group, and some prior findings from that study are replicated, albeit with a very small effect size. It neglects much of the rest of the relevant literature however, e.g. in the numerous unreferenced claims in the Introduction.

Following the reviewers’ recommendations, we provide more literature in the introduction section, such that all claims are now backed-up by references and that the current study appears less based exclusively on our previous neuroimaging study.

In particular, we now provide references for the following claims:

– p.3: “Previous animal research suggested that wanting and liking relate to dissociable neurochemical mechanisms (Berridge and Kringelbach, 2015; Berridge and Valenstein, 1991)”

– p.3:“Both dopaminergic and opioidergic neurons project to reward circuits in the striatum as well as to the prefrontal cortex (Delay-Goyet, Zajac, Javoy-Agid, Agid, and Roques, 1987; Lidow, Goldman-Rakic, Gallager, and Rakic, 1991)”

– p.4: “In particular, the ventral striatum encodes the currently behaviorally relevant reward dimension and dynamically switches functional connectivity with wanting- and liking-encoding prefrontal regions accordingly (Weber, Kahnt, Quednow, and Tobler, 2018; switching between frontostriatal loops according to behavioral relevance constitutes one possibility how dopamine (and by extension opioids) can affect functions involving prefrontal cortex and the striatum: Cools, 2011). This is also in line with the view that wanting and liking are preferentially represented in partially distinct subregions of the striatum (which are in turn connected to different input and output regions; Peciña (2008)).”

– p.4: “We hypothesized that, on a behavioral level, opioid receptor blockade will reduce both wanting and liking (Buchel, Miedl, and Sprenger, 2018; Chelnokova et al., 2014; Eikemo et al., 2016), whereas reduced dopaminergic neurotransmission will selectively affect wanting (Cawley et al., 2013; Korb et al., 2020; Venugopalan et al., 2011)”

Methods2) Please specify the urine tox screen used. Please explain how participants were recruited and identified as being free of exclusion criteria. For example, was a formal, semi-structured clinical interview for psychiatric disorders conducted?

We now clarify which urine drug test we used in our study (p.20):

“A qualitative drug urine screening test (M-10/5-DT, Diagnostik Nord, Schwerin, Germany) was performed to rule out illicit drug use prior the test session (amphetamines, barbiturates, buprenorphine, benzodiazepines, cannabis, cocaine, MDMA, methadone and morphine/opiates).”

Regarding participant recruitment, we clarify that participants were recruited via email from the internal participant pool of the Laboratory for Social and Neural Systems Research, which includes mainly students from the University of Zurich and ETH Zurich (p.19):

“A total of 121 healthy volunteers (58 females; Mage = 21.8 years, range = 18-30), recruited via email from the internal pool of the Laboratory for Social and Neural Systems Research (which includes mainly students from the University of Zurich and ETH Zurich), participated in the study.”

The exclusion criteria were checked by a paper-pencil questionnaire rather than by a formal clinical interview. Participants had to indicate whether they had been diagnosed with a psychiatric disorder in the past. We clarify the procedure on p.19:

“Participants were screened prior to participation for exclusion criteria, including a history of brain disease or injury, surgery to the head or heart and neurological or psychiatric diseases (including alcohol use disorder, depression, schizophrenia, bipolar disorders, claustrophobia or Parkinson symptoms) via paper-pencil questionnaires.”

3) The authors write that wanting and liking ratings were elicited twice (before and after fMRI), yet analyses appear to indicate that these judgements were also elicited during fMRI? A number of other details about the task are missing, such as the rationale for selection of items, whether these were actually won or hypothetically won, why the 'difficulty' was set to 50%, and why only explicit ratings were included.

We apologize for the lack of clarity here. Wanting and liking ratings were collected during the acquisition of fMRI data in the scanner. There were two fMRI sessions. One before and one after a game in which participants won half of the items. We now formulate this more clearly in the manuscript on p.5 (see below). In addition, we clarify that participants actually won the items in the game, i.e., they could take the won items home after the experiment. We therefore selected everyday items that should (at least to some extent) be wanted and liked by the majority of our participants stemming from the student population in Zurich. The “difficulty” of the game was set to 50% in order to have equal numbers of won and lost items for the statistical analysis. We clarify these issues on p.5:

“We collected wanting and liking ratings for all items in the MRI scanner twice, once before (pre-test session) and once after (post-test session) participants played a game on the computer where they won or lost 50% of the items (in order to have equal numbers of won and lost items for the statistical analysis). […] We therefore selected everyday items (e.g., batteries or candles – for the full list of items, see Weber et al. (2018)) that should be both wanted and liked by the majority of our participants from the Zurich student population.”

Lastly, concerning the inclusion of only explicit self-report measures for wanting and liking, we followed our previous study (Weber et al., 2018) and decided to avoid the issue that implicit measures like face muscle activity are arguably harder to interpret than explicit measures. Having said this, we also recognize the possibility that implicit measures might be more sensitive to drug effects than explicit measures of wanting and liking (see Korb et al., 2020). We therefore extended the discussion of this issue on p.16:

“In line with our previous study (Weber et al., 2018), we had decided to use self-report ratings to avoid the issue that implicit measures of liking, such as face muscle activity are more open to alternative interpretations (Pool, Sennwald, Delplanque, Brosch, and Sander, 2016), but we acknowledge that implicit measures might be more sensitive to pharmacological interventions.”

4) The authors report that they did not orthogonalize the parametric modulators modelling the wanting and liking ratings. While this makes sense as a way to ensure that each modulator captures unique variance, presumably wanting and liking ratings are tightly correlated. Was that the case? Could the authors provide some details about the average correlation between the ratings?

In the revised manuscript on p.23, we now report the mean item-specific correlation between wanting and liking ratings, which was r = 0.71.

“The mean item-specific correlation between wanting and liking ratings was r = 0.71. We did not orthogonalize the parametric modulators, such that the results for the regressors indicate the unique variance explained by wanting or liking ratings.”

5) The reviewers appreciated the power analysis. However some details are missing so that the numbers provided can be fully appreciated by the readers. In particular, the study cited as a reference for effect sizes (Kahnt et al., 2015) used a different task and different analyses, which makes it unclear how the authors were able to derive a relevant effect size estimate. Also, since there are many effects that can be tested in the present study (i.e. various behavioral and brain-related effects and correlations), it is important to report explicitly for which of these effects sample size was optimized.

In the revised manuscript, we provide more details regarding the power analysis. For the current power analysis, we had used the impact of amisulpride (versus placebo) on the representation of reward in the OFC reported on page 4107 in the study by Kahnt et al. (2015). However, we agree with the reviewers that the current study used different tasks and additional pharmacological interventions (naltrexone) compared with the Kahnt et al. study. While power analyses assume that a study tries to replicate a previously observed effect, the novelty of our investigation made it impossible to conduct a proper power analysis including all effects on the behavioral and neural level. Instead, we decided to collect a sample size that should be sufficient to observe pharmacological effects on neural reward presentations, but we acknowledge that this represents a best guess for the true required sample size rather than a proper power analysis based on a replication of previous effects.

We discuss this issue in the revised manuscript on p.19:

“According to power analysis assuming the effect size (Cohen’s d = 0.65) from a previous study in our lab on the impact of amisulpride on value representations in the neural reward system (Kahnt, Weber, Haker, Robbins, and Tobler, 2015), 38 participants per group allow detecting a significant effect (α = 5%) with a power of 80%. […] However, given the differences between the current study design and the study by Kahnt et al. (2015), we note that the power might not have been optimal for all statistical tests in the current investigation (e.g., drug effects on explicit ratings or functional connectivity).”

6) The reviewers also appreciated the pre-registration on clinicaltrials.gov. However again some details seem to be missing. The pre-registration mentions a PIT task and a delay discounting task, but no rating task. This discrepancy requires explanation. Furthermore, the pre-registration implies that this study was embedded in a larger project involving more tasks than the one reported here. It is important to report this information explicitly, and explain how the rating task took place in the context of this larger protocol.

We thank the reviewer for pointing to this important issue. The current study was indeed part of a larger project where we had investigated also the effects of naltrexone and amisulpride on a PIT task and a delay discounting task. These two tasks were administered outside the scanner, after the wanting/liking rating task in the scanner, such that they cannot have affected performance in our rating task. The results for the PIT and delay discounting tasks have been published (Weber et al., 2016, Translational Psychiatry). Re-checking the preregistration on clinicaltrials.gov, we realized that indeed only the PIT and delay discounting tasks had been preregistered, not the rating task. We apologize for this mistake and now clarify in the revised manuscript that the current sub-project was not mentioned in the pre-registration (p.20).

“The study was approved by the ethics committee of the canton of Zurich and was part of a larger project where we investigated also pharmacological effects on Pavlovian-to-instrumental transfer and delay discounting (published in Weber et al. (2016)). These tasks were administered after the rating task reported in the current manuscript. The larger project (though not the rating task) was preregistered on www.clinicaltrials.gov (NCT02557984).”

Results7) The reviewers queried the statistics presented in the analysis of the behavioural data. In Figure 1, the amisulpride group shows clearly elevated ratings of wanting and liking, at a magnitude similar to the effect of naltrexone on wanting, but the difference between between amisulpride and placebo is reported as non-significant for liking ratings. The statistics reported do not appear to match the means – e.g. in panel D. The difference between naltrexone and placebo for wanting looks similar (with even larger between-subject variance), so it seems surprising that the effect of amisulpride on liking ratings is non-significant. Can the authors verify there is not a mistake? It also seems from Figure 1D that amisulpride might have a main (increasing) effect on ratings: have the authors checked for such a main effect?

We thank the reviewers for these important comments. We checked the statistics and found them to be correct as reported in the manuscript. However, we also understand the reviewers’ impression that e.g. Figure 1D suggests an influence of amisulpride on liking, even though this effect is not significant in the statistics. This can be explained by the fact that Figure 1 shows the mean wanting and liking ratings for each participant (collapsed across items), while the mixed linear models statistically analyze ratings for each item and participant (with participant-specific and item-specific random intercepts). The means and standard errors in the figures might therefore occasionally give an impression that deviates from the actual statistical results. To avoid misunderstanding, we added a statement to the legend for Figure 1 clarifying that the figure displays the mean wanting and liking ratings for illustration purpose, while the statistical analyses are based on participant- and item-specific ratings.

“For illustration purposes, participant-specific mean wanting/liking ratings are plotted on a scale from 0 to 10, while the statistical analyses are conducted on the participant- and item-specific wanting and liking ratings.”

8) The authors should explain how the degrees of freedom reported for the contrasts were determined – they often do not appear to match either the number of participants or the number of observations.

We now clarify that degrees of freedom were computed using the Satterthwaite approximation (p.22), which is the default option in the lmerTest package. We note that with the Satterthwaite approximation the degrees of freedom depend on the variance within samples, such that degrees of freedom can differ even for tests including the same number of observations or participants.

“Degrees of freedom and p-values were computed using the Satterthwaite approximation with the lmerTest package.”

9) The description of the Judgement × Item type interaction as "marginally significant" is inappropriate. A quick check with http://statcheck.io/ shows that the two-tailed p-value is 0.11, so the one-tailed p-value is 0.055. Following recent recommendations in the field (e.g. Gibbs and Gibbs, 2015; Otte et al., 2021), especially in a context where some scientists have called for lowering the statistical threshold of p<0.05 for significance (Benjamin et al., 2017), the authors should report this result as non-significant, and/or substantiate it with a complementary statistical approach.

We apologize for the lack of clarity here. It is correct that the precise two-tailed p-value is 0.1087 (0.054 one-tailed). We used the term “marginally significant” to indicate that the p-value is close to the statistical threshold (other fields, e.g., economics use p=0.1), but following the reviewers’ recommendations we now refer to this effect as non-significant trend-level effect (p.6):

“we observed a non-significant but trend-level effect for the *Judgement* × *Item type* interaction, β = 0.50, *t*(111) = 1.61, *p* = 0.054, one-tailed, suggesting that winning versus losing items tended to have dissociable effects on wanting versus liking of the items”.

10) On a related point, on p5 a p-value of 0.52 cannot support a claim of the form "wanting ratings did not differ between won and lost items". Such a p-value > 0.05 merely suggests that the data at hand don't provide sufficient evidence to reject the null. Bayesian statistics could potentially be used to make the sort of claims made by the authors. The same issue applies on p. 6: "amisulpride did not change wanting ratings relative to placebo, β = -1.39, t(116) = 0.22, p = 0.83".

We apologize for these unclear formulations. We re-formulated these sentences and clarify that non-significant results represent only absence of evidence for meaningful differences rather than evidence for the null hypotheses. We agree that in principle Bayesian statistics could be used to corroborate claims in favor of the null hypothesis, but as we did not mean to make such claims we opted to formulate more carefully.

p.6:

“Separate analyses for wanting and liking ratings revealed no significant difference in wanting ratings between won and lost items, β = 0.29, *t*(115) = 0.64, *p* = 0.52”.

p.6:

“whereas amisulpride (mean = 5.2, sd = 1.7) showed no significant effects on wanting ratings relative to placebo, β = -1.39, *t*(116) = 0.22, *p* = 0.83.”

11) Ideally one would like to establish a causal mechanism from the present results: is reduced wanting following opioid receptor blockade mediated by a direct action of naltrexone on frontostriatal connectivity (as suggested in the title of this submission)? Have the authors have considered performing a mediation-style analysis?

We thank the reviewers for this interesting suggestion. We conducted a mediation analysis by adding the individual wanting-related DLPFC-striatum connectivity strength during wanting judgements as additional predictor to the behavioral model assessing the impact of naltrexone on wanting. When we assessed the significance of the indirect path (i.e., whether naltrexone effects on wanting can statistically be explained via modulation of DLPFC-striatum connectivity) using the mediation package in R, the indirect path showed only a trend-level effect, p = 0.06. Thus, the data provide no clear evidence that reduced wanting after opioid receptor blockade can causally be explained by naltrexone effects on frontostriatal connectivity. We therefore weakened our statements throughout the manuscript (including the title; see below) to avoid the impression that the naltrexone effect on behavior can causally be explained by changes in frontostriatal connectivity.

We report the mediation analysis on p.12:

“The observed drug effects on the neural level raise the question whether the impact of naltrexone on behavioral wanting ratings can statistically be explained by its influence on DLPFC-striatum connectivity. […] We must therefore be careful with interpreting the naltrexone effects on brain connectivity as the potential cause for the behavioral drug effects.”

We added a discussion of the non-significant mediation analysis on p.15:

“because we observed no significant brain-behavior mediation effect (p = 0.06), the stronger DLPFC-striatum connectivity under naltrexone should not be interpreted as the cause for changes in behavioral ratings under naltrexone.”

12) On p9 it would informative to present the striatal activation observed for the correlation with relevant wanting ratings. The statistics are reported for only one unilateral peak voxel, so it is hard to assess how robust this result is. This is particularly important since this result is supposed to be a replication of the authors' previous work (Weber et al., 2018).

In the revised manuscript, we present the wanting-related striatum activation during wanting judgements in Figure 2C and we also report more details for the wanting-related striatum activation. We clarify that the activation does not survive FWE-correction at the voxel-wise whole-brain level (only FWE-correction at the cluster level), suggesting that wanting-related striatum activation might be weaker in the current study compared with our previous work. However, given the clear a priori hypothesis based on our previous findings, it is justified to use SVC to test for wanting-related striatum activity. We report the stats for the whole-brain correction on p.11:

“We note that wanting-related striatal activation did not survive FWE-correction at the whole-brain level (*p* = 0.14, although it did survive whole-brain FWE-correction at the cluster level, p < 0.001), such that this effect appears to be somewhat weaker than in our previous study (Weber et al., 2018).”

13) The authors note that DLPFC-striatum connectivity was associated with self-reported wanting and that naltrexone increased this connectivity further. Since naltrexone also decreased self-reported wanting, are these results contradictory? This needs clarification.

Thank you for this interesting comment. One possibility for reconciling these two findings, in line with previous research (van den Bos et al., 2014), is that the DLPFC exerts top-down (inhibitory) influence over the striatum to reduce impulsivity. Assuming that impulsivity is associated with higher levels of wanting, such an influence may be important only for highly wanted items, whereas there is no need for an inhibition of striatal wanting signals for less desired items. From this perspective, the enhanced wanting-related DLPFC-striatum connectivity under naltrexone can be interpreted as stronger prefrontal top-down control over the striatum particularly for highly wanted items (i.e., when top-down impulse control is actually required). Because the study followed a between-group design, we cannot determine the strength of the pharmacological intervention on the individual level and it is thus not possible to test whether the naltrexone effects on behavior and brain activity are correlated. Moreover, as described above, also the mediation analysis showed no significant effect (p=0.06). In the added discussion of top-down control on p.15, we are thus cautious with interpreting the stronger functional connectivity under naltrexone as the cause for the naltrexone effects on behavioral wanting ratings:

“According to this view, the positive relationship between wanting ratings and DLPFC-striatum connectivity might indicate that DLPFC exerts top-down control over striatal representations of wanting predominantly for highly wanted items, whereas there may be less need for inhibitory top-down control for less desired goods (van den Bos et al., 2014). In any case, because we observed no significant brain-behavior mediation effect, the stronger DLPFC-striatum connectivity under naltrexone should not be interpreted as the cause for changes in behavioral ratings under naltrexone.”

Discussion14) The authors used an intermediate dose of amisulpride (400 mg), which has been suggested to exert a mixture of pre- and post-synaptic effects depending on the participant, with pre-synaptic blockage likely to increase D1 receptor activation. Therefore any group-wise effect may cancel out across participants (see e.g. van der Schaaf et al., 2012; Sescousse et al., 2016; Eisenegger et al., 2014). Could this be a reason for the limited effects of amisulpride observed in the present study?

We agree with the reviewers that amisulpride can have both presynaptic and postsynaptic effects, and that these effects might cancel out on the group level, which might contribute to the weak to absent effects of amisulpride in the current study. In this case the effects of amisulpride should vary as a function of body weight as proxy of effective dose. However, adding body weight as predictor to the behavioral MGLMs did not change the overall result pattern, the impact of naltrexone remained significant, while we observed no significant effects of amisulpride. We report this analysis on p.7:

“Amisulpride can show both pre-synaptic and post-synaptic effects depending on the administered dose. […] There was thus no evidence for dose-dependent effects of amisulpride on wanting or liking ratings.”

Moreover, we note that in other studies using the same dose of amisulpride and a similar delay between drug intake and task performance we did observe significant amisulpride effects on decision-making (Burke et al., 2018; Soutschek et al., 2017). This further speaks against the possibility that the weak effects of amisulpride in the rating task can be explained by the administered dose. We elaborated the discussion of the non-significant amisulpride effects on p.17-18:

“Lastly, while high doses of amisulpride (≥400 mg) reduce postsynaptic dopaminergic signaling, lower doses of amisulpride increase dopaminergic activity via presynaptic mechanisms (Schoemaker et al., 1997). […] However, contrary to this view, we observed no significant amisulpride effects even when controlling for body weight as proxy for effective dose, and we note that in previous studies we had observed effects of 400 mg amisulpirde on behavior (Burke et al., 2018; Soutschek et al., 2017) and multivariate neural data (Kahnt et al., 2015). It seems thus unlikely that the null effects of amisulpride on the rating task can be explained solely by the chosen dosage.”

15) Some of the key results appear to be unilateral, i.e. specifically in the left hemisphere (striatal activity scaling with wanting ratings and DLPFC-striatum connectivity). Such a unilateral effect could reflect limited statistical power and fragile results, but also a truly lateralized effect. It would be worth commenting on this in the Discussion.

Good point. In the literature, reward-related functional connectivity with the striatum was reported for both the left and the right DLPFC. In agreement with this notion, a direct comparison between left and right hemisphere revealed no significant difference. We therefore avoid any claims regarding lateralization in the current manuscript (p.15).

“It is worth noting that in the current study we observed significant effects predominantly in the left hemisphere. […] We therefore do not make any claims regarding whether this result pattern represents just a power issue or a truly lateralized effect.”

16) Based on previous literature it is surprising that the striatum was not part of the neural correlates of wanting and liking (Figure 2) – striatal activation is only identified when the ratings are restricted to the relevant ones. Yet, based on studies like Barta et al. (2013) or Lebreton et al. (2009) that show that the striatum is at the core of a robust and automatic valuation system, one might expect activity in this region to correlated with ratings even when these are not explicitly required. Could the authors comment on this?

We agree with the reviewers that there is evidence that the striatum is part of an automatic valuation system. While GLM-1 (modelling wanting and liking ratings irrespective of their relevance) did not show significant wanting-related striatum activation, it is important to keep in mind that the parametric modulators in GLM-1 were not orthogonalized, explaining thus only the unique contributions of wanting and liking to differences in neural activity. In fact, when we computed a further GLM where liking was orthogonalized with respect to wanting, then wanting ratings significantly correlated with activation in the striatum, with the peak in the left striatum (*z* = 7.09, whole-brain FWE-corrected, *p* < 0.001, peak = [-9 11 -1]) but the cluster extended to both hemispheres. Thus, our findings are consistent with the well-described role of the striatum for reward processing, though this had not been evident in the GLM with non-orthogonalized parametric modulators due to the shared variance between wanting and liking.

We now report the results for the GLM with orthogonalized parametric modulators on p.8-9:

“GLM-1 revealed no significant wanting- or liking-related striatal activation, which may appear surprising given the canonical role of the striatum for reward processing (Bartra, McGuire, and Kable, 2013). […] In fact, when we orthogonalized liking with respect to wanting (such that the regressor for wanting explained the variance shared by wanting and liking), we observed bilateral wanting-related activation in the striatum (z = 7.09, whole-brain FWE-corrected, p < 0.001, peak = [-9 11 -1]).”

References:

Berridge, K. C., and Kringelbach, M. L. (2015). Pleasure systems in the brain. Neuron, 86(3), 646-664. doi:10.1016/j.neuron.2015.02.018

Berridge, K. C., and Valenstein, E. S. (1991). What psychological process mediates feeding evoked by electrical stimulation of the lateral hypothalamus? Behav Neurosci, 105(1), 3-14. doi:10.1037//0735-7044.105.1.3

Buchel, C., Miedl, S., and Sprenger, C. (2018). Hedonic processing in humans is mediated by an opioidergic mechanism in a mesocorticolimbic system. *eLife*, 7. doi:10.7554/*eLife*.39648

Cawley, E. I., Park, S., aan het Rot, M., Sancton, K., Benkelfat, C., Young, S. N.,... Leyton, M. (2013). Dopamine and light: dissecting effects on mood and motivational states in women with subsyndromal seasonal affective disorder. J Psychiatry Neurosci, 38(6), 388-397. doi:10.1503/jpn.120181

Chelnokova, O., Laeng, B., Eikemo, M., Riegels, J., Loseth, G., Maurud, H.,... Leknes, S. (2014). Rewards of beauty: the opioid system mediates social motivation in humans. Mol Psychiatry, 19(7), 746-747. doi:10.1038/mp.2014.1

Cools, R. (2011). Dopaminergic control of the striatum for high-level cognition. Curr Opin Neurobiol, 21(3), 402-407. doi:10.1016/j.conb.2011.04.002

Delay-Goyet, P., Zajac, J.-M., Javoy-Agid, F., Agid, Y., and Roques, B. (1987). Regional distribution of μ, δ and κ opioid receptors in human brains from controls and parkinsonian subjects. Brain Res, 414(1), 8-14.

Eikemo, M., Loseth, G. E., Johnstone, T., Gjerstad, J., Willoch, F., and Leknes, S. (2016). Sweet taste pleasantness is modulated by morphine and naltrexone. Psychopharmacology (Berl), 233(21-22), 3711-3723. doi:10.1007/s00213-016-4403-x

Kahnt, T., Weber, S. C., Haker, H., Robbins, T. W., and Tobler, P. N. (2015). Dopamine D2-receptor blockade enhances decoding of prefrontal signals in humans. J Neurosci, 35(9), 4104-4111. doi:10.1523/JNEUROSCI.4182-14.2015

Korb, S., Gotzendorfer, S. J., Massaccesi, C., Sezen, P., Graf, I., Willeit, M.,... Silani, G. (2020). Dopaminergic and opioidergic regulation during anticipation and consumption of social and nonsocial rewards. e*Life*, 9. doi:10.7554/*eLife*.55797

Lidow, M. S., Goldman-Rakic, P. S., Gallager, D., and Rakic, P. (1991). Distribution of dopaminergic receptors in the primate cerebral cortex: quantitative autoradiographic analysis using [3H] raclopride,[3H] spiperone and [3H] SCH23390. Neuroscience, 40(3), 657-671.

Pool, E., Sennwald, V., Delplanque, S., Brosch, T., and Sander, D. (2016). Measuring wanting and liking from animals to humans: A systematic review. Neurosci Biobehav Rev, 63, 124-142. doi:10.1016/j.neubiorev.2016.01.006

Venugopalan, V. V., Casey, K. F., O'Hara, C., O'Loughlin, J., Benkelfat, C., Fellows, L. K., and Leyton, M. (2011). Acute phenylalanine/tyrosine depletion reduces motivation to smoke cigarettes across stages of addiction. Neuropsychopharmacology, 36(12), 2469-2476. doi:10.1038/npp.2011.135

Weber, S. C., Beck-Schimmer, B., Kajdi, M. E., Muller, D., Tobler, P. N., and Quednow, B. B. (2016). Dopamine D2/3- and mu-opioid receptor antagonists reduce cue-induced responding and reward impulsivity in humans. Transl Psychiatry, 6(7), e850. doi:10.1038/tp.2016.113

Weber, S. C., Kahnt, T., Quednow, B. B., and Tobler, P. N. (2018). Frontostriatal pathways gate processing of behaviorally relevant reward dimensions. PLoS Biol, 16(10), e2005722. doi:10.1371/journal.pbio.2005722

[Editors' note: further revisions were suggested prior to acceptance, as described below.]

The reviewers and editors feel that the manuscript has been improved but there are several remaining major issues that need to be addressed, as outlined below. Please note that if these issues are not satisfactorily addressed in your revised submission then unfortunately we will not be able to consider the manuscript further, as it is not editorial practice to issue multiple revise resubmit decisions at eLife.1) The most important issue is that there remains a discrepancy between similar effect sizes of naltrexone/amisulpride on wanting (albeit in opposite directions) and the corresponding pattern of P-values obtained from the hierarchical analysis. A similar issue is also present in relation to liking ratings. The authors need to explore this discrepancy in considerably more detail and resolve it, as follows:a) Conduct a non-hierarchical analysis using the mean ratings for wanting and liking (in two separate models, one for wanting, one for liking). The reason for this is that it appears from the data depicted in Figure 1D that amisulpride may increase wanting and also liking (where the effect may actually be even greater). The reviewers noted that for liking ratings the mean difference is ~0.5 points and the SDs are actualy lower than for wanting ratings at 1.0/1.2, which is suggestive of a larger effect than the effect of naltrexone on wanting which is significant in the hierarchical model. It would also be useful to provide the standardised effect sizes (Cohen's d) for the 4 comparisons against placebo (2 for wanting, 2 for liking).

Following the reviewers’ advice, we conducted a non-hierarchical analysis of pharmacological effects on wanting and liking ratings. For this, we computed the mean wanting and liking ratings across all items, separately for each participant and session (pre-test versus post-test). The analysis of wanting ratings replicated the significant main effect of naltrexone versus placebo (*p* = 0.03), while amisulpride showed no significant effect on mean wanting ratings (p = 0.82). Mean liking ratings were neither affected by naltrexone (p = 0.98) nor amisulpride (p = 0.23). Thus, also a non-hierarchical analysis of aggregated mean data provides no evidence for significant drug effects. We currently did not include it in the manuscript as it just replicates the results from the hierarchical analysis; however, we would be happy to add it to the manuscript if the reviewers prefer us to do so. As suggested by the reviewer, we now report Cohen’s d for the drug effects on wanting and liking on p.6-7:

“Judgement type-specific analyses suggested that wanting ratings were significantly reduced under naltrexone (mean = 4.5, standard deviation (sd) = 1.0) relative to placebo (mean = 4.9, sd = 1.0), β = -13.85, *t*(115) = 2.12, *p* = 0.04, Cohen’s d = 0.47, whereas amisulpride (mean = 4.9, sd = 1.0) showed no significant effects on wanting ratings relative to placebo, β = -1.39, *t*(116) = 0.22, *p* = 0.83, Cohen’s d = 0.05. […] Taken together, our findings provide evidence for involvement of opioidergic neurotransmission in wanting judgements.”

Still, this leaves open the question how the perceived discrepancy between the figure and the statistics can be explained. We carefully checked our scripts for creating the plots and, embarrassingly, discovered an error in the calculation of the mean ratings used for the figures (wrong assignment of individual values to drug groups in the plotting script). We now corrected the figures as well as the means/standard deviations reported in the manuscript, and in our view the discrepancy between figures and statistics has now resolved. We apologize for this confusion and are grateful to the reviewers whose careful inspection of the figures allowed us to correct this mistake.

b) Assuming that the above analyses using the mean ratings provide a discrepant pattern of significance to the hierarchical analysis, this then needs to be investigated thoroughly and explained in the manuscript, both for wanting and for liking ratings. The authors need to dig into the data carefully and figure out why this discrepancy arises. For example, if amisulpride makes participants more variable in their ratings (or naltrexone make them more consistent), this would be important for interpretation. Or perhaps some assumptions of the hierarchical model have been broken? Or perhaps the covariance structure requires amending? Or perhaps the model did not converge? Without resolution of this discrepancy it will not be possible to consider the manuscript further.

Given that the non-hierarchical analysis shows converging results with the hierarchical analysis and we corrected the error in the script for the plots, there is no need for further digging into the data as far as we can see.

2) The authors now report the magnitude of the correlation between wanting and liking ratings, which is unsurprisingly high (r = 0.71). Since they have not serially orthogonalized the parametric regressors in the main analysis, this means that much of the variance of these ratings is simply removed. For this reason it is not clear how much we can infer from the non-significant drug effects, considering that these were assessed using only a fraction of the ratings variance, which may result in an insensitive analysis. Therefore further analyses are required here to substantiate the conclusion that there were no drug effects (as reported on the top of p10 – it is assumed that currently this refers to the model without serial orthogonalisation, although this should be stated explicitly for clarity).The authors do provide some results from an analysis using serial orthogonalisation, with liking orthogonalised against wanting (p9), yielding the expected striatal activation for the parametric effect of wanting (which then carries the shared variance), which is reassuring – as they note this suggests that the striatal signal is substantively affected by the colinearity between wanting and liking ratings. Please additionally report the drug effects in this analysis. The authors should also report the effects from an analysis in which the serial orthogonalisation is performed in the alternate order (i.e. wanting against liking, such that the liking regressor now carries the shared variance), including both the main parametric effect (this time of liking) and drug effects.

We thank the reviewers for this suggestion. As recommended, we now report the results for two further GLMs, one where liking was orthogonalized against wanting, and one where wanting was orthogonalized against liking. These GLMs suggest that both wanting and liking ratings correlate with activity in the neural reward system (striatum, VMPFC, and PCC). In addition, we also tested for drug effects on the neural correlates of wanting or liking, but no effect survived correction for multiple comparisons.

In the revised manuscript, we report these analyses on p.9-10:

“GLM-1 revealed no significant wanting- or liking-related striatal activation, which may appear surprising given the canonical role of the striatum for reward processing (Bartra, McGuire, and Kable, 2013). […] However, also in the GLMs with orthogonalized parametric modulators, we observed no effects of naltrexone or amisulpride (relative to placebo) on wanting (GLM-3) or liking (GLM-4) ratings even at lenient statistical thresholds (*p* < 0.001 uncorrected, cluster size > 20 voxels).”

We describe these models in the Materials and methods section on p.25-26:

“Finally, we computed two further models, one where the liking regressor in GLM-1 was orthogonalized with respect to wanting (such that the regressor for wanting contained the variance shared by wanting and liking; GLM-3) and one where wanting was orthogonalized with respect to liking (GLM-4).”

3) It is helpful that the authors added the information that previous data from the same study were published in a 2016 paper by Weber et al. Oddly they do not mention the results of that paper, even in the discussion of the (apparent – see point 3 below) non-significant effects of amisulpride. The 2016 findings are highly relevant, since the amisulpride group was found to suppress cue-based responding and reward impulsivity. Similar results, but weaker, were reported for naltrexone. Both groups also reported lower mood than the placebo group.The authors explain that the PIT and delay discounting tasks were completed after the end of scanning, i.e. after 60 minutes absorption time + 90 minutes fmri rating task = minimum 2.5 hours after drug administration. Hence, it seems highly relevant for the interpretation of the present data that, in the exact same participants, the same dose of amisulpride reported to show a null during 1-2.5 hours, showed what are (presumably) expected effects after 2.5 hours. Therefore it is necessary to mention this prior publication from the same study in the Introduction, and discuss the results, especially with respect to dose timing, in the Discussion.

We agree with the reviewers that the results of the already published data inform the interpretation of the current findings. As recommended, we now mention that the study was part of a larger project already in the introduction section (p.4):

“This study was part of a larger project investigating also the roles of opioidergic and dopaminergic activity for reward impulsivity (Weber et al., 2016).”

When discussing the non-significant amisulpride effects in the Discussion section, we now clarify that in the same participants amisulpride showed significant effects in other tasks. We discuss that this discrepancy might be explained either by different sensitivities of the tasks to dopaminergic manipulations or by the time course of amisulpride effects (p.17-18).

“We note though that in the same sample of participants amisulpride showed significant effects on tasks for cue reactivity and delay discounting (Weber et al., 2016), which were administered 2.5 hours after drug intake (while the rating task started 1 hour after drug intake). Due to this difference in timing, it is not possible to decide whether the different amisulpride effects on these tasks can be explained by different sensitivities of these tasks to dopaminergic manipulations or by the time course of amisulpride effects.”